# DRIVE: One-bit Distributed Mean Estimation

**Shay Vargaftik** *
VMware Research
shayv@vmware.com

**Ran Ben Basat** *
University College London
r.benbasat@cs.ucl.ac.uk

**Amit Portnoy** *
Ben-Gurion University
amitport@post.bgu.ac.il

**Gal Mendelson**
Stanford University
galmen@stanford.edu

**Yaniv Ben-Itzhak**
VMware Research
ybenitzhak@vmware.com

**Michael Mitzenmacher**
Harvard University
michaelm@eecs.harvard.edu

## Abstract

We consider the problem where $n$ clients transmit $d$-dimensional real-valued vectors using $d(1 + o(1))$ bits each, in a manner that allows the receiver to approximately reconstruct their mean. Such compression problems naturally arise in distributed and federated learning. We provide novel mathematical results and derive computationally efficient algorithms that are more accurate than previous compression techniques. We evaluate our methods on a collection of distributed and federated learning tasks, using a variety of datasets, and show a consistent improvement over the state of the art.

## 1 Introduction

In many computational settings, one wishes to transmit a $d$-dimensional real-valued vector. For example, in distributed and federated learning scenarios, multiple participants (a.k.a. *clients*) in distributed SGD send gradients to a parameter server that averages them and updates the model parameters accordingly [1]. In these applications and others (e.g., traditional machine learning methods such K-Means and power iteration [2] or other methods such as geometric monitoring [3]), sending *approximations* of vectors may suffice. Moreover, the vectors' dimension $d$ is often large (e.g., in neural networks, $d$ can exceed a billion [4, 5, 6]), so sending compressed vectors is appealing.

Indeed, recent works have studied how to send vector approximations using representations that use a small number of bits per entry (e.g., [2, 7, 8, 9, 10, 11]). Further, recent work has shown direct training time reduction from compressing the vectors to one bit per coordinate [12]. Most relevant to our work are solutions that address the distributed mean estimation problem. For example, [2] uses the randomized Hadamard transform followed by stochastic quantization (a.k.a. randomized rounding). When each of the $n$ clients transmits $O(d)$ bits, their Normalized Mean Squared Error (NMSE) is bounded by $O\left(\frac{\log d}{n}\right)$. They also show a $O(\frac{1}{n})$ bound with $O(d)$ bits via variable-length encoding, albeit at a higher computational cost. The sampling method of [10] yields an $O\left(\frac{r \cdot R}{n}\right)$ NMSE bound using $d(1 + o(1))$ bits *in expectation*, where $r$ is each coordinate's representation length and $R$ is the normalized average variance of the sent vectors. Recently, researchers proposed to use Kashin's representation [11, 13, 14]. Broadly speaking, it allows representing a $d$-dimensional vector in (a higher) dimension $\lambda \cdot d$ for some $\lambda > 1$ using small coefficients. This results in an $O\left(\frac{\lambda^2}{(\sqrt{\lambda}-1)^4 \cdot n}\right)$ NMSE bound, where each client transmits $\lambda \cdot d(1 + o(1))$ bits [14]. A recent work [15] suggested an algorithm where if all clients' vectors have pairwise distances of at most $y \in \mathbb{R}$ (i.e., for any client pair $\mathfrak{c}_1, \mathfrak{c}_2$, it holds that $\left\| x_{(\mathfrak{c}_1)} - x_{(\mathfrak{c}_2)} \right\|_2 \leqslant y$), the resulting MSE is $O(y^2)$ (which is tight with respect to

---

*Equal Contribution.

35th Conference on Neural Information Processing Systems (NeurIPS 2021).

y) using $O(1)$ bits per coordinate on average. This solution provides a stronger MSE bound when vectors are sufficiently close (and thus $y$ is small) but does not improve the worst-case guarantee.

We step back and focus on approximating $d$-dimensional vectors using $d(1 + o(1))$ bits (e.g., one bit per dimension and a lower order overhead). We develop novel biased and unbiased compression techniques based on (uniform as well as structured) random rotations in high-dimensional spheres. Intuitively, after a rotation, the coordinates are identically distributed, allowing us to estimate each coordinate with respect to the resulting distribution. Our algorithms do not require expensive operations, such as variable-length encoding or computing the Kashin's representation, and are fast and easy to implement. We obtain an $O\left(\frac{1}{n}\right)$ NMSE bound using $d(1 + o(1))$ bits, regardless of the coordinates' representation length, improving over previous works. Evaluation results indicate that this translates to a consistent improvement over the state of the art in different distributed and federated learning tasks.

## 2   Problem Formulation and Notation

**1b - Vector Estimation.** We start by formally defining the *1b - vector estimation* problem. A sender, called Buffy, gets a real-valued vector $x \in \mathbb{R}^d$ and sends it using a $d(1 + o(1))$ bits message (i.e., asymptotically *one bit per coordinate*). The receiver, called Angel, uses the message to derive an estimate $\widehat{x}$ of the original vector $x$. We are interested in the quantity $\|x - \widehat{x}\|_2^2$, which is the sum of squared errors (SSE), and its expected value, the Mean Squared Error (MSE). For ease of exposition, we hereafter assume that $x \neq 0$ as this special case can be handled with one additional bit. Our goal is to minimize the *vector*-NMSE (denoted *vNMSE*), defined as the normalized MSE, i.e., $\frac{\mathbb{E}\left[\|x - \widehat{x}\|_2^2\right]}{\|x\|_2^2}$.

**1b - Distributed Mean Estimation.** The above problem naturally generalizes to the *1b - Distributed Mean Estimation* problem. Here, we have a set of $n \in \mathbb{N}$ *clients* and a *server*. Each client $\mathfrak{c} \in \{1, \dots, n\}$ has its own vector $x_{(\mathfrak{c})} \in \mathbb{R}^d$, which it sends using a $d(1+o(1))$-bits message to the server. The server then produces an estimate $\widehat{x_{\mathrm{avg}}} \in \mathbb{R}^d$ of the average $x_{\mathrm{avg}} = \frac{1}{n} \sum_{\mathfrak{c}=1}^n x_{(\mathfrak{c})}$ with the goal of minimizing its *NMSE*, defined as the average estimate's MSE normalized by the average norm of the clients' original vectors, i.e., $\frac{\mathbb{E}\left[\|x_{\mathrm{avg}} - \widehat{x_{\mathrm{avg}}}\|_2^2\right]}{\frac{1}{n} \cdot \sum_{\mathfrak{c}=1}^n \|x_{(\mathfrak{c})}\|_2^2}$.

**Notation.** We use the following notation and definitions throughout the paper:

*Subscripts.* $x_i$ denotes the $i$'th *coordinate* of the vector $x$, to distinguish it from client $\mathfrak{c}$'s vector $x_{(\mathfrak{c})}$.

*Binary-sign.* For a vector $x \in \mathbb{R}^d$, we denote its binary-sign function as $\mathrm{sign}(x)$, where $\mathrm{sign}(x)_i = 1$ if $x_i \geqslant 0$ and $\mathrm{sign}(x)_i = -1$ if $x_i < 0$.

*Unit vector.* For any (non-zero) real-valued vector $x \in \mathbb{R}^d$, we denote its normalized vector by $\breve{x} = \frac{x}{\|x\|_2}$. That is, $\breve{x}$ and $x$ has the same direction and it holds that $\|\breve{x}\|_2 = 1$.

*Rotation Matrix.* A matrix $R \in \mathbb{R}^{d \times d}$ is a rotation matrix if $R^T R = I$. The set of all rotation matrices is denoted as $\mathcal{O}(d)$. It follows that $\forall R \in \mathcal{O}(d) : det(R) \in \{-1, 1\}$ and $\forall x \in \mathbb{R}^d : \|x\|_2 = \|Rx\|_2$.

*Random Rotation.* A random rotation $\mathcal{R}$ is a distribution over all random rotations in $\mathcal{O}(d)$. For ease of exposition, we abuse the notation and given $x \in \mathbb{R}^d$ denote the random rotation of $x$ by $\mathcal{R}(x) = Rx$, where $R$ is drawn from $\mathcal{R}$. Similarly, $\mathcal{R}^{-1}(x) = R^{-1}x = R^T x$ is the inverse rotation.

*Rotation Property.* A quantity that determines the guarantees of our algorithms is $\mathcal{L}_{\mathcal{R},x}^d = \frac{\|\mathcal{R}(\breve{x})\|_1^2}{d}$ (note the use of the $L_1$ norm). We show that rotations with high $\mathcal{L}_{\mathcal{R},x}^d$ values yield better estimates.

**Shared Randomness.** We assume that Buffy and Angel have access to shared randomness, e.g., by agreeing on a common PRNG seed. Shared randomness is studied both in communication complexity (e.g., [16]) and in communication reduction in machine learning systems (e.g., [2, 7]). In our context, it means that Buffy and Angel can generate the same random rotations without communication.

## 3   The DRIVE Algorithm

We start by presenting DRIVE (Deterministically RoundIng randomly rotated VEctors), a novel 1b - Vector Estimation algorithm. Later, we extend DRIVE to the 1b - Distributed Mean Estimation

**Algorithm 1** DRIVE

**Buffy:**
1: Compute $\mathcal{R}(x)$, $S$.
2: Send $\big(S, \text{sign}(\mathcal{R}(x))\big)$ to Angel.

**Angel:**
1: Compute $\widehat{\mathcal{R}(x)} = S \cdot \text{sign}\big(\mathcal{R}(x)\big)$.
2: Estimate $\widehat{x} = \mathcal{R}^{-1}\big(\widehat{\mathcal{R}(x)}\big)$.

---

problem. In DRIVE, Buffy uses shared randomness to sample a rotation matrix $R \sim \mathcal{R}$ and rotates the vector $x \in \mathbb{R}^d$ by computing $\mathcal{R}(x) = Rx$. Buffy then calculates $S$, a scalar quantity we explain below. Buffy then sends $\big(S, \text{sign}(\mathcal{R}(x))\big)$ to Angel. As we discuss later, sending $\big(S, \text{sign}(\mathcal{R}(x))\big)$ requires $d(1 + o(1))$ bits. In turn, Angel computes $\widehat{\mathcal{R}(x)} = S \cdot \text{sign}(\mathcal{R}(x)) \in \{-S, +S\}^d$. It then uses the shared randomness to generate the same rotation matrix and employs the inverse rotation, i.e., estimates $\widehat{x} = \mathcal{R}^{-1}(\widehat{\mathcal{R}(x)})$. The pseudocode of DRIVE appears in Algorithm 1.

The properties of DRIVE depend on the rotation $\mathcal{R}$ and the *scale parameter* $S$. We consider both uniform rotations, that provide stronger guarantees, and structured rotations that are orders of magnitude faster to compute. As for the scale $S = S(x, R)$, its exact formula determines the characteristics of DRIVE's estimate, e.g., having minimal vNMSE or being unbiased. The latter allows us to apply DRIVE to the 1b - Distributed Mean Estimation (Section 4.2) and get an NMSE that decreases proportionally to the number of clients.

We now prove a general result on the SSE of DRIVE that applies to any random rotation $\mathcal{R}$ and any vector $x \in \mathbb{R}^d$. In the following sections, we use this result to obtain the vNMSE when considering specific rotations and scaling methods as well as analyzing their guarantees.

**Theorem 1.** *The SSE of DRIVE is:* $\|x - \widehat{x}\|_2^2 = \|x\|_2^2 - 2 \cdot S \cdot \|\mathcal{R}(x)\|_1 + d \cdot S^2$.

*Proof.* The SSE in estimating $\mathcal{R}(x)$ using $\widehat{\mathcal{R}(x)}$ equals that of estimating $x$ using $\widehat{x}$. Therefore,

$$\|x - \widehat{x}\|_2^2 = \|\mathcal{R}(x - \widehat{x})\|_2^2 = \|\mathcal{R}(x) - \mathcal{R}(\widehat{x})\|_2^2 = \left\|\mathcal{R}(x) - \widehat{\mathcal{R}(x)}\right\|_2^2$$

$$= \|\mathcal{R}(x)\|_2^2 - 2\left\langle \mathcal{R}(x), \widehat{\mathcal{R}(x)}\right\rangle + \left\|\widehat{\mathcal{R}(x)}\right\|_2^2 = \|x\|_2^2 - 2\left\langle \mathcal{R}(x), \widehat{\mathcal{R}(x)}\right\rangle + \left\|\widehat{\mathcal{R}(x)}\right\|_2^2. \quad (1)$$

Next, we have that,

$$\left\langle \mathcal{R}(x), \widehat{\mathcal{R}(x)}\right\rangle = \sum_{i=1}^d \mathcal{R}(x)_i \cdot \widehat{\mathcal{R}(x)}_i = S \cdot \sum_{i=1}^d \mathcal{R}(x)_i \cdot \text{sign}\big(\mathcal{R}(x)_i\big) = S \cdot \|\mathcal{R}(x)\|_1, \quad (2)$$

$$\left\|\widehat{\mathcal{R}(x)}\right\|_2^2 = \sum_{i=1}^d \widehat{\mathcal{R}(x)}_i^2 = d \cdot S^2. \quad (3)$$

Substituting Eq. (2) and Eq. (3) in Eq. (1) yields the result. $\qquad\square$

## 4 DRIVE With a Uniform Random Rotation

We first consider the thoroughly studied uniform random rotation (e.g., [17, 18, 19, 20, 21]), which we denote by $\mathcal{R}_U$. The sampled matrix is denoted by $R_U \sim \mathcal{R}_U$, that is, $\mathcal{R}_U(x) = R_U \cdot x$. An appealing property of a uniform random rotation is that, as we show later, it admits a scaling that results in a low constant vNMSE even with unbiased estimates.

### 4.1 1b - Vector Estimation

Using Theorem 1, we obtain the following result. The result holds for any rotation, including $\mathcal{R}_U$.

**Lemma 1.** *For any $x \in \mathbb{R}^d$, DRIVE's SSE is minimized by $S = \frac{\|\mathcal{R}(x)\|_1}{d}$ (that is, $S = \frac{\|Rx\|_1}{d}$ is determined after $R \sim \mathcal{R}$ is sampled). This yields a vNMSE of $\frac{\mathbb{E}\big[\|x - \widehat{x}\|_2^2\big]}{\|x\|_2^2} = 1 - \mathbb{E}\big[\mathcal{L}_{\mathcal{R},x}^d\big]$.*

*Proof.* By Theorem 1, to minimize the SSE we require

$$\frac{\partial}{\partial S}\big(\|x\|_2^2 - 2 \cdot S \cdot \|\mathcal{R}(x)\|_1 + d \cdot S^2\big) = -2 \cdot \|\mathcal{R}(x)\|_1 + 2 \cdot d \cdot S = 0,$$

leading to $S = \frac{\|\mathcal{R}(x)\|_1}{d}$. Then, the SSE of DRIVE becomes:

$$\|x - \widehat{x}\|_2^2 = \|x\|_2^2 - 2 \cdot S \cdot \|\mathcal{R}(x)\|_1 + d \cdot S^2 = \|x\|_2^2 - 2 \cdot \frac{\|\mathcal{R}(x)\|_1^2}{d} + d \cdot \frac{\|\mathcal{R}(x)\|_1^2}{d^2}$$
$$= \|x\|_2^2 - \frac{\|\mathcal{R}(x)\|_1^2}{d} = \|x\|_2^2 - \frac{\|x\|_2^2 \cdot \|\mathcal{R}(\check{x})\|_1^2}{d} = \|x\|_2^2 \left(1 - \frac{\|\mathcal{R}(\check{x})\|_1^2}{d}\right).$$

Thus, the normalized SSE is $\frac{\|x - \widehat{x}\|_2^2}{\|x\|_2^2} = 1 - \mathcal{L}_{\mathcal{R},x}^d$. Taking expectation yields the result. $\square$

Interestingly, for the uniform random rotation, $\mathcal{L}_{\mathcal{R}_U,x}^d$ follows the same distribution for all $x$. This is because, by the definition of $\mathcal{R}_U$, it holds that $\mathcal{R}_U(\check{x})$ is distributed uniformly over the unit sphere for any $x$. Therefore DRIVE's vNMSE depends only on the dimension $d$. We next analyze the vNMSE attainable by the best possible $S$, as given in Lemma 1, when the algorithm uses $\mathcal{R}_U$ and is not required to be unbiased. In particular, we state the following theorem whose proof appears in Appendix A.1 (all appendices appear in the Supplementary Material and the extended paper version [22]).

**Theorem 2.** *For any $x \in \mathbb{R}^d$, the vNMSE of DRIVE with $S = \frac{\|\mathcal{R}_U(x)\|_1}{d}$ is $\left(1 - \frac{2}{\pi}\right)\left(1 - \frac{1}{d}\right)$.*

### 4.2 1b - Distributed Mean Estimation

An appealing property of DRIVE with a uniform random rotation, established in this section, is that with a proper scaling parameter $S$, the estimate is unbiased. That is, for any $x \in \mathbb{R}^d$, our scale guarantees that $\mathbb{E}[\widehat{x}] = x$. Unbiasedness is useful when generalizing to the Distributed Mean Estimation problem. Intuitively, when $n$ clients send their vectors, any biased algorithm would result in an NMSE that may not decrease with respect to $n$. For example, if they have the same input vector, the bias would remain after averaging. Instead, an unbiased encoding algorithm has the property that when all clients act (e.g., use different PRNG seeds) independently, the NMSE decreases proportionally to $\frac{1}{n}$.

Another useful property of uniform random rotation is that its distribution is unchanged when composed with other rotations. We use it in the following theorem's proof, given in Appendix A.2.

**Theorem 3.** *For any $x \in \mathbb{R}^d$, set $S = \frac{\|x\|_2^2}{\|\mathcal{R}_U(x)\|_1}$. Then DRIVE satisfies $\mathbb{E}[\widehat{x}] = x$.*

Now, we proceed to obtain vNMSE guarantees for DRIVE's unbiased estimate.

**Lemma 2.** *For any $x \in \mathbb{R}^d$, DRIVE with $S = \frac{\|x\|_2^2}{\|\mathcal{R}_U(x)\|_1}$ has a vNMSE of $\mathbb{E}\left[\frac{1}{\mathcal{L}_{\mathcal{R}_U,x}^d}\right] - 1$.*

*Proof.* By Theorem 1, the SSE of the algorithm satisfies:

$$\|x\|_2^2 - 2 \cdot S \cdot \|R_U \cdot x\|_1 + d \cdot S^2 = \|x\|_2^2 - 2 \cdot \frac{\|x\|_2^2}{\|R_U \cdot x\|_1} \cdot \|R_U \cdot x\|_1 + d \cdot \left(\frac{\|x\|_2^2}{\|R_U \cdot x\|_1}\right)^2$$
$$= d \cdot \left(\frac{\|x\|_2^2}{\|x\|_2 \|R_U \cdot \check{x}\|_1}\right)^2 - \|x\|_2^2 = d \cdot \frac{\|x\|_2^2}{\|R_U \cdot \check{x}\|_1^2} - \|x\|_2^2 = \|x\|_2^2 \cdot \left(\left(\frac{d}{\|R_U \cdot \check{x}\|_1^2}\right) - 1\right).$$

Normalizing by $\|x\|_2^2$ and taking expectation over $R_U$ concludes the proof. $\square$

Our goal is to derive an upper bound on the above expression and thus upper-bound the vNMSE. Most importantly, we show that even though the estimate is unbiased and we use only a single bit per coordinate, the vNMSE does not increase with the dimension and is bounded by a small constant. In particular, in Appendix A.3, we prove the following:

**Theorem 4.** *For any $x \in \mathbb{R}^d$, the vNMSE of DRIVE with $S = \frac{\|x\|_2^2}{\|\mathcal{R}_U(x)\|_1}$ satisfies:*

*(i) For all $d \geqslant 2$, it is at most 2.92. (ii) For all $d \geqslant 135$, it is at most $\frac{\pi}{2} - 1 + \sqrt{\frac{(6\pi^3 - 12\pi^2) \cdot \ln d + 1}{d}}$.*

This theorem yields strong bounds on the vNMSE. For example, the vNMSE is lower than 1 for $d \geqslant 4096$ and lower than 0.673 for $d \geqslant 10^5$. Finally, we obtain the following corollary,

**Corollary 1.** *For any $x \in \mathbb{R}^d$, the vNMSE tends to $\frac{\pi}{2} - 1 \approx 0.571$ as $d \to \infty$.*

Recall that DRIVE's above scale $S$ is a function of both $x$ and the sampled $R_U$. An alternative approach is to *deterministically* set $S$ to $\frac{\|x\|_2^2}{\mathbb{E}\left[\|\mathcal{R}_U(x)\|_1\right]}$. As we prove in Appendix A.4, the resulting scale is $\frac{\|x\|_2 \cdot (d-1) \cdot \mathrm{B}(\frac{1}{2}, \frac{d-1}{2})}{2d}$, where B is the Beta function. Interestingly, this scale no longer depends on $x$ but only on its norm. In the appendix, we also prove that the resulting vNMSE is bounded by $\frac{\pi}{2} - 1$ *for any $d$*. In practice, we find that the benefit is marginal.

Finally, with a vNMSE guarantee for the unbiased estimate by DRIVE, we obtain the following key result for the 1b - Distributed Mean Estimation problem, whose proof appears in Appendix A.5. We note that this result guarantees (e.g., see [23]) that distributed SGD, where the participants' gradients are compressed with DRIVE, converges at the same asymptotic rate as without compression.

**Theorem 5.** *Assume $n$ clients, each with its own vector $x_{(\mathfrak{c})} \in \mathbb{R}^d$. Let each client independently sample $R_{U,\mathfrak{c}} \sim \mathcal{R}_U$ and set its scale to $\frac{\|x_{(\mathfrak{c})}\|_2^2}{\|R_{U,\mathfrak{c}} \cdot x_{(\mathfrak{c})}\|_1}$. Then, the server average estimate's NMSE satisfies:* $\frac{\mathbb{E}\left[\|x_{avg} - \widehat{x_{avg}}\|_2^2\right]}{\frac{1}{n} \cdot \sum_{\mathfrak{c}=1}^{n} \|x_{(\mathfrak{c})}\|_2^2} = \frac{vNMSE}{n}$, *where vNMSE is given by Lemma 2 and is bounded by Theorem 4.*

To the best of our knowledge, DRIVE is the first algorithm with a provable NMSE of $O(\frac{1}{n})$ for the 1b - Distributed Mean Estimation problem (i.e., with $d(1 + o(1))$ bits). In practice, we use only $d + O(1)$ bits to implement DRIVE. We use the $d(1 + o(1))$ notation to ensure compatibility with the theoretical results; see Appendix B for a discussion.

# 5 Reducing the vNMSE with DRIVE$^+$

To reduce the vNMSE further, we introduce the DRIVE$^+$ algorithm. In DRIVE$^+$, we also use a scale parameter, denoted $S^+ = S^+(x, R)$ to differentiate it from the scale $S$ of DRIVE. Here, instead of reconstructing the rotated vector in a symmetric manner, i.e., $\widehat{\mathcal{R}(x)} \in S \cdot \{-1, 1\}^d$, we have that $\widehat{\mathcal{R}(x)} \in S^+ \cdot \{c_1, c_2\}^d$ where $c_1, c_2$ are computed using K-Means clustering with $K = 2$ over the $d$ entries of the rotated vector $\mathcal{R}(x)$. That is, $c_1, c_2$ are chosen to minimize the SSE over any choice of two values. This does not increase the (asymptotic) time complexity over the random rotations considered in this paper as solving K-Means for the special case of one-dimensional data is deterministically solvable in $O(d \log d)$ (e.g., [24]). Notice that DRIVE$^+$ still requires $d(1 + o(1))$ bits as we communicate $(S^+ \cdot c_1, S^+ \cdot c_2)$ and a single bit per coordinate, indicating its nearest centroid. We defer the pseudocode and analyses of DRIVE$^+$ to Appendix C. We show that with proper scaling, for both the 1b - Vector Estimation and 1b - Distributed Mean Estimation problems, DRIVE$^+$ yields guarantees that are at least as strong as those of DRIVE.

# 6 DRIVE with a Structured Random Rotation

Uniform random rotation generation usually relies on QR factorization (e.g., see [25]), which requires $O(d^3)$ time and $O(d^2)$ space. Therefore, uniform random rotation can only be used in practice to rotate low-dimensional vectors. This is impractical for neural network architectures with many millions of parameters. To that end, we continue to analyze DRIVE and DRIVE$^+$ with the (randomized) Hadamard transform, a.k.a. *structured* random rotation [2, 26], that admits a fast *in-place*, parallelizable, $O(d \log d)$ time implementation [27, 28, 29]. We start with a few definitions.

**Definition 1.** *The Walsh-Hadamard matrix ([30]) $H_{2^k} \in \{+1, -1\}^{2^k \times 2^k}$ is recursively defined via:*
$H_{2^k} = \begin{pmatrix} H_{2^{k-1}} & H_{2^{k-1}} \\ H_{2^{k-1}} & -H_{2^{k-1}} \end{pmatrix}$ *and $H_1 = (1)$. Also, $(\frac{1}{\sqrt{d}}H) \cdot (\frac{1}{\sqrt{d}}H)^T = I$ and $\det(\frac{1}{\sqrt{d}}H) \in [-1, 1]$.*

**Definition 2.** *Let $R_H$ denote the rotation matrix $\frac{HD}{\sqrt{d}} \in \mathbb{R}^{d \times d}$, where $H$ is a Walsh-Hadamard matrix and $D$ is a diagonal matrix whose diagonal entries are i.i.d. Rademacher random variables (i.e., taking values uniformly in $\pm 1$). Then $\mathcal{R}_H(x) = R_H \cdot x = \frac{1}{\sqrt{d}}H \cdot (x_1 \cdot D_{11}, \ldots, x_d \cdot D_{dd})^T$ is the randomized Hadamard transform of $x$ and $\mathcal{R}_H^{-1}(x) = R_H^T \cdot x = \frac{DH}{\sqrt{d}} \cdot x$ is the inverse transform.*

## 6.1 1b - Vector Estimation

Recall that the vNMSE of DRIVE, when minimized using $S = \frac{\|\mathcal{R}(x)\|_1}{d}$, is $1 - \mathbb{E}\left[\mathcal{L}_{\mathcal{R},x}^d\right]$ (see Lemma 1). We now bound this quantity of DRIVE with a structured random rotation.

**Lemma 3.** *For any dimension $d \geqslant 2$ and vector $x \in \mathbb{R}^d$, the vNMSE of DRIVE with a structured random rotation and scale $S = \frac{\|\mathcal{R}_H(x)\|_1}{d}$ is: $1 - \mathbb{E}\left[\mathcal{L}_{\mathcal{R}_H,x}^d\right] \leqslant \frac{1}{2}$.*

*Proof.* Observe that for all $i$, $\mathbb{E}\left[|\mathcal{R}_H(x)_i|\right] = \mathbb{E}\left[\left|\sum_{j=1}^d \frac{x_j}{\sqrt{d}} H_{ij} D_{jj}\right|\right]$. Since $\{H_{ij} D_{jj} \mid j \in [d]\}$ are i.i.d. Rademacher random variables we can use the Khintchine inequality [31, 32] which implies that $\frac{1}{\sqrt{2d}} \cdot \|x\|_2 \leqslant \mathbb{E}\left[|\mathcal{R}_H(x)_i|\right] \leqslant \frac{1}{\sqrt{d}} \cdot \|x\|_2$ (see [33, 34] for simplified proofs). We conclude that:

$$\mathbb{E}\left[\mathcal{L}_{\mathcal{R}_H,x}^d\right] = \frac{1}{d} \cdot \mathbb{E}\left[\|\mathcal{R}_H(\breve{x})\|_1^2\right] \geqslant \frac{1}{d} \cdot \mathbb{E}\left[\|\mathcal{R}_H(\breve{x})\|_1\right]^2 \geqslant \frac{1}{d} \cdot \left(\sum_{i=1}^d \frac{1}{\sqrt{2d}}\right)^2 = \frac{1}{2} .$$

This bound is sharp since for $d \geqslant 2$ we have that $\mathcal{L}_{\mathcal{R}_H,x}^d = \frac{1}{2}$ for $x = (\frac{1}{\sqrt{2}}, \frac{1}{\sqrt{2}}, 0, \ldots, 0)^T$. $\qquad\square$

Observe that unlike for the uniform random rotation, $\mathbb{E}\left[\mathcal{L}_{\mathcal{R}_H,x}^d\right]$ depends on $x$. We also note that this bound of $\frac{1}{2}$ applies to DRIVE$^+$ (with scale $S^+ = 1$) as we show in Appendix C.2.

## 6.2 1b - Distributed Mean Estimation

For an arbitrary $x \in \mathbb{R}^d$ and $\mathcal{R}$, and in particular for $\mathcal{R}_H$, the estimates of DRIVE cannot be made unbiased. For example, for $x = (\frac{2}{3}, \frac{1}{3})^T$ we have that $\text{sign}(\mathcal{R}_H(x)) = (D_{11}, D_{11})^T$ and thus $\widehat{\mathcal{R}_H(x)} = S \cdot (D_{11}, D_{11})^T$. This implies that $\widehat{x} = \mathcal{R}_H^{-1}(\widehat{\mathcal{R}_H(x)}) = \frac{1}{\sqrt{2}} \cdot D \cdot H \cdot S \cdot (D_{11}, D_{11})^T = \sqrt{2} \cdot S \cdot D \cdot (D_{11}, 0)^T = \sqrt{2} \cdot S \cdot (D_{11}^2, 0)^T = (\sqrt{2} \cdot S, 0)^T$. Therefore, $\mathbb{E}[\widehat{x}] \neq x$ regardless of the scale.

Nevertheless, we next provide evidence for why when the input vector is high dimensional and admits finite moments, a structured random rotation performs similarly to a uniform random rotation, yielding all the appealing aforementioned properties. Indeed, it is a common observation that the distribution of machine learning workloads and, in particular, neural network gradients are governed by such distributions (e.g., lognormal [35] or normal [36, 37]).

We seek to show that at high dimensions, the distribution of $\mathcal{R}_H(x)$ is sufficiently similar to that of $\mathcal{R}_U(x)$. By definition, the distribution of $\mathcal{R}_U(x)$ is that of a uniformly at random distributed point on a sphere. Previous studies of this distribution for high dimensions (e.g., [38, 39, 40, 41]) have shown that individual coordinates of $\mathcal{R}_U(x)$ converge to the same normal distribution and that these coordinates are "weakly" dependent in the sense that the joint distribution of every $O(1)$-sized subset of coordinates is similar to that of independent normal variables for large $d$.

We hereafter assume that $x = (x_1, \ldots, x_d)$, where the $x_i$s are i.i.d. and that $\mathbb{E}[x_j^2] = \sigma^2$ and $\mathbb{E}[|x_j|^3] = \rho < \infty$ for all $j$. We show that $\mathcal{R}_H(x)_i$ converges to the same normal distribution for all $i$. Let $F_{i,d}(x)$ be the cumulative distribution function (CDF) of $\frac{1}{\sigma} \cdot \mathcal{R}_H(x)_i$ and $\Phi$ be the CDF of the standard normal distribution. The following lemma, proven in Appendix D.1, shows the convergence.

**Lemma 4.** *For all $i$, $\mathcal{R}_H(x)_i$ converges to a normal variable: $\sup_{x \in \mathbb{R}} |F_{i,d}(x) - \Phi(x)| \leqslant \frac{0.409 \cdot \rho}{\sigma^3 \sqrt{d}}$.*

With this result, we continue to lay out evidence for the "weak dependency" among the coordinates. We do so by calculating the moments of their joint distribution in increasing subset sizes showing that these moments converge to those of independent normal variables. Previous work has shown that a structured random rotation on vectors with specific distributions results in "weakly dependent" normal variables. This line of research [42, 43, 44] utilized the Hadamard transform for a different purpose. Their goal was to develop a computationally cheap method to generate independent normally distributed variables from simpler (e.g., uniform) distributions. We apply their analysis to our setting.

We partially rely on the following observation that the Hadamard matrix satisfies.

**Observation 1.** *([42]) The Hadamard product (coordinate-wise product), $H_{\langle i \rangle} \circ H_{\langle \ell \rangle}$, of two rows $H_{\langle i \rangle}, H_{\langle \ell \rangle}$ in the Hadamard matrix yields another row at the matrix $H_{\langle i \rangle} \circ H_{\langle \ell \rangle} = H_{\langle 1 + (i-1) \oplus (\ell-1) \rangle}$. Here, $(i-1) \oplus (\ell-1)$ is the bitwise xor of the $(\log d)$-sized binary representation of $(i-1)$ and $(\ell-1)$. It follows that $\sum_{j=1}^d H_{ij} H_{\ell j} = \sum_{j=1}^d (H_{\langle i \rangle} \circ H_{\langle \ell \rangle})_j = \sum_{j=1}^d H_{1+(i-1) \oplus (\ell-1), j}$.*

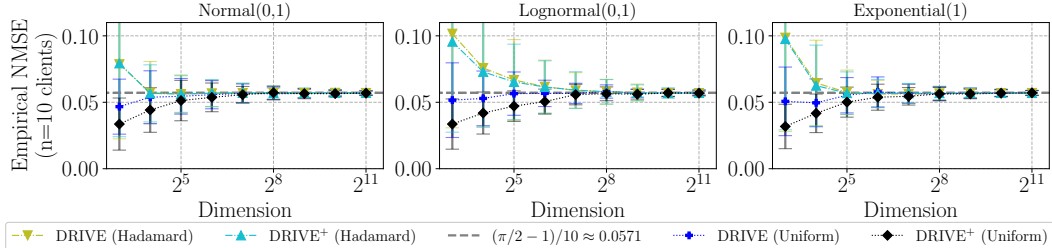

Figure 1: Distributed mean estimation comparison: each data point is averaged over $10^4$ trials. In each trial, *the same* (randomly sampled) vector is sent by $n = 10$ clients.

We now analyze the moments of the rotated variables, starting with the following observation. It follows from the sign-symmetry of $D$ and matches the joint distribution of i.i.d. normal variables.

**Observation 2.** *All odd moments containing $\mathcal{R}_H(x)$ entries are 0. That is,*
$$\forall q \in \mathbb{N}, \forall i_1, \ldots, i_{2q+1} \in \{1, \ldots, d\} : \mathbb{E}\left[\mathcal{R}_H(x)_{i_1} \cdot \ldots \cdot \mathcal{R}_H(x)_{i_{2q+1}}\right] = 0.$$

Therefore, we need to examine only even moments. We start with showing that the second moments also match with the distribution of independent normal variables.

**Lemma 5.** *For all $i \neq \ell$ it holds that $\mathbb{E}\left[(\mathcal{R}_H \cdot x)_i \cdot (\mathcal{R}_H \cdot x)_\ell\right] = 0$, whereas $\mathbb{E}\left[(\mathcal{R}_H \cdot x)_i^2\right] = \sigma^2$.*

*Proof.* Since $\{D_{jj} \mid j \in \{1, \ldots, d\}\}$ are sign-symmetric and i.i.d., $\mathbb{E}\left[(\mathcal{R}_H \cdot x)_i \cdot (\mathcal{R}_H \cdot x)_\ell\right] = \mathbb{E}\left[\frac{1}{d}(\sum_{j=1}^d x_j H_{ij} D_{jj}) \cdot (\sum_{j=1}^d x_j H_{\ell j} D_{jj})\right] = \mathbb{E}\left[x_j^2\right] \cdot \frac{1}{d} \cdot \sum_{j=1}^d H_{ij} H_{\ell j}$. Notice that $\sum_{j=1}^d H_{1j} = d$ and $\sum_{j=1}^d H_{ij} = 0$ for all $i > 1$. Thus, by Observation 1 we get 0 if $i \neq \ell$ and $\sigma^2$ otherwise. $\square$

We have established that the coordinates are pairwise uncorrelated. Similar but more involved analysis shows that the same trend continues under the assumption of the existence of $x$'s higher moments. In Appendix D.2 we analyze the 4th moments showing that they indeed approach the 4th moments of independent normal variables with a rate of $\frac{1}{d}$; the reader is referred to [42] for further intuition and higher moments analysis. We therefore expect that using DRIVE and DRIVE$^+$ with Hadamard transform will yield similar results to that of a uniform random rotation at high dimensions and when the input vectors respect the finite moments assumption.

In addition to the theoretical evidence, in Figure 1, we show experimental results comparing the measured NMSE for the 1b - Distributed Mean Estimation problem with $n = 10$ clients (all given the *same* vector so biases do not cancel out) for DRIVE and DRIVE$^+$ using both uniform and structured random rotations over three different distributions. The results indicate that all variants yield similar NMSEs in reasonable dimensions, in line with the theoretical guarantee of Corollary 1 and Theorem 5.

## 7 Evaluation

We evaluate DRIVE and DRIVE$^+$, comparing them to standard and recent state-of-the-art techniques. We consider classic distributed learning tasks as well as federated learning tasks (e.g., where the data distribution is not i.i.d. and clients may change over time). All the distributed tasks are implemented over PyTorch [45] and all the federated tasks are implemented over TensorFlow Federated [46]. We focus our comparison on vector quantization algorithms and recent sketching techniques and exclude sparsification methods (e.g., [47, 48, 49, 50]) and methods that involve client-side memory since these can often work in conjunction with our algorithms.

**Datasets.** We use MNIST [51, 52], EMNIST [53], CIFAR-10 and CIFAR-100 [54] for image classification tasks; a next-character-prediction task using the Shakespeare dataset [55]; and a next-word-prediction task using the Stack Overflow dataset [56]. Additional details appear in Appendix E.1.

**Algorithms.** Since our focus is on the distributed mean estimation problem and its federated and distributed learning applications, we run DRIVE and DRIVE$^+$ with the unbiased scale quantities.[1]

We compare against several alternative algorithms: (1) *FedAvg* [1] that uses the full vectors (i.e., each coordinate is represented using a 32-bit float); (2) Hadamard transform followed by 1-bit stochastic

| Dimension ($d$) | Hadamard + 1-bit SQ | Kashin + 1-bit SQ | Drive (Uniform) | Drive$^+$ (Uniform) | Drive (Hadamard) | Drive$^+$ (Hadamard) |
|---|---|---|---|---|---|---|
| 128 | 0.5308, *0.34* | 0.2550, *2.12* | 0.0567, *40.4* | **0.0547**, *40.7* | 0.0591, *0.36* | 0.0591, *0.72* |
| 8,192 | 1.3338, *0.57* | 0.3180, *3.42* | **0.0571**, *5088* | **0.0571**, *5101* | **0.0571**, *0.60* | **0.0571**, *1.06* |
| 524,288 | 2.1456, *0.79* | 0.3178, *4.69* | — | — | **0.0571**, *0.82* | **0.0571**, *1.35* |
| 33,554,432 | 2.9332, *27.1* | 0.3179, *332* | — | — | **0.0571**, *27.2* | **0.0571**, *37.8* |

Table 1: Empirical NMSE and average per-vector encoding time (in milliseconds, on an RTX 3090 GPU) for distributed mean estimation with $n = 10$ clients (same as in Figure 1) and Lognormal(0,1) distribution. Each entry is a (NMSE, *time*) tuple and the most accurate result is highlighted in **bold**.

quantization (SQ) [2, 10]; (3) Kashin's representation followed by 1-bit stochastic quantization [11]; (4) *TernGrad* [8], which clips coordinates larger than 2.5 times the standard deviation, then performs 1-bit stochastic quantization on the absolute values and separately sends their signs and the maximum coordinate for scale (we note that TernGrad is a low-bit variant of a well-known algorithm called *QSGD* [9], and we use TernGrad since we found it to perform better in our experiments);[2] and (5-6) *Sketched-SGD* [57] and *FetchSGD* [58], which are both count-sketch [59] based algorithms designed for distributed and federated learning, respectively.

We note that Hadamard with 1-bit stochastic quantization is our most fair comparison, as it uses the same number of bits as DRIVE$^+$ (and slightly more than DRIVE) and has similar computational costs. This contrasts with Kashin's representation, where both the number of bits and the computational costs are higher. For example, a standard TensorFlow Federated implementation (e.g., see "CLASS KASHINHADAMARDENCODINGSTAGE" hyperparameters at [60]) uses a minimum of 1.17 bits per coordinate, and three iterations of the algorithm resulting in five Hadamard transforms for each vector. Also, note that TernGrad uses an extra bit per coordinate for sending the sign. Moreover, the clipping performed by TernGrad is a heuristic procedure, which is orthogonal to our work.

For each task, we use a subset of datasets and the most relevant competition. Detailed configuration information and additional results appear in Appendix E. We first evaluate the vNMSE-Speed tradeoffs and then proceed to federated and distributed learning experiments.

**vNMSE-Speed Tradeoff.** Appearing in Table 1, the results show that our algorithms offer the lowest NMSE and that the gap increases with the dimension. As expected, DRIVE and DRIVE$^+$ with uniform rotation are more accurate for small dimensions but are significantly slower. Similarly, DRIVE is as accurate as DRIVE$^+$, and both are significantly more accurate than Kashin (by a factor of 4.4×-5.5×) and Hadamard (9.3×-51×) with stochastic quantization. Additionally, DRIVE is 5.7×-12× faster than Kashin and about as fast as Hadamard. In Appendix E.3 we discuss the results, give the complete experiment specification, and provide measurements on a commodity machine.

We note that the above techniques, including DRIVE, are more computationally expensive than linear-time solutions like TernGrad. Nevertheless, DRIVE's computational overhead becomes insignificant for modern learning tasks. For example, our measurements suggest that it can take 470 ms for computing the gradient on a ResNet18 architecture (for CIFAR100, batch size = 128, using NVIDIA GeForceGTX 1060 (6GB) GPU) while the encoding of DRIVE (Hadamard) takes 2.8 ms. That is, the overall computation time is only increased by 0.6% while the error reduces significantly. Taking the transmission and model update times into consideration would reduce the importance of the compression time further.

**Federated Learning.** We evaluate over four tasks: (1) EMNIST over customized CNN architecture with two convolutional layers with $\approx 1.2M$ parameters [61]; (2) CIFAR-100 over ResNet-18 [54] ; (3) a next-character-prediction task using the Shakespeare dataset [1]; (4) a next-word-prediction task using the Stack Overflow dataset [62]. Both (3) and (4) use LSTM recurrent models [63] with $\approx 820K$ and $\approx 4M$ parameters, respectively. We use code, client partitioning, models, hyperparameters, and validation metrics from the federated learning benchmark of [62].

---

[1]For DRIVE the scale is $S = \frac{\|x\|_2^2}{\|\mathcal{R}(x)\|_1}$ (see Theorem 3). For DRIVE$^+$ the scale is $S^+ = \frac{\|x\|_2^2}{\|c\|_2^2}$, where $c \in \{c_1, c_2\}^d$ is the vector indicating the nearest centroid to each coordinate in $\mathcal{R}(x)$ (see Section 5).

[2]When restricted to two quantization levels, TernGrad is identical to QSGD's max normalization variant with clipping (slightly better due to the ability to represent 0).

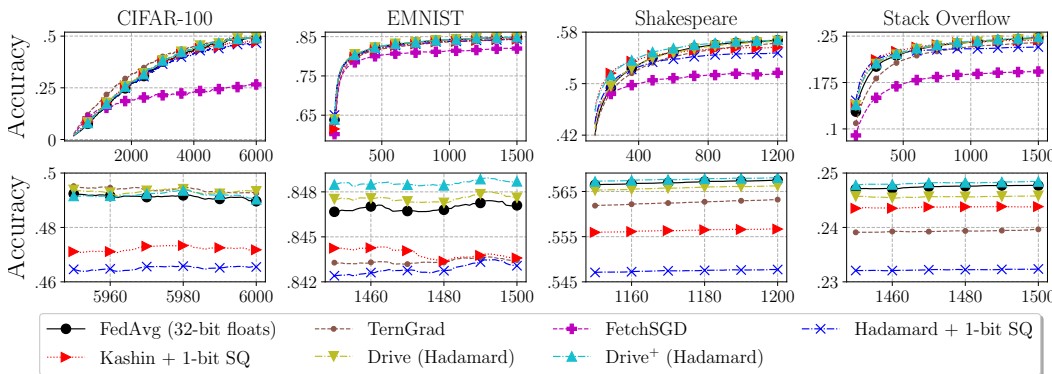

Figure 2: Accuracy per round on various federated learning tasks. Smoothing is done using a rolling mean with a window size of 150. The second row zooms-in on the last 50 rounds.

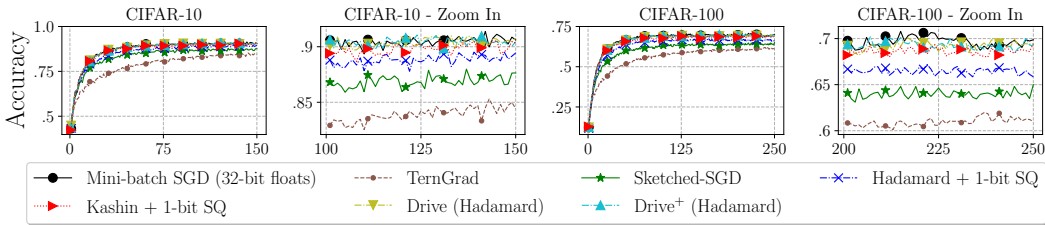

Figure 3: Accuracy per round on distributed learning tasks, with a zoom-in on the last 50 rounds.

The results are depicted in Figure 2. We observe that in all tasks, DRIVE and DRIVE$^+$ have accuracy that is competitive with that of the baseline, FedAvg. In CIFAR-100, TernGrad and DRIVE provide the best accuracy. For the other tasks, DRIVE and DRIVE$^+$ have the best accuracy, while the best alternative is either Kashin + 1-bit SQ or TernGrad, depending on the task. Hadamard + 1-bit SQ, which is the most similar to our algorithms (in terms of both bandwidth and compute), provides lower accuracy in all tasks. Additional details and hyperparameter configurations are presented in Appendix E.4.

**Distributed CNN Training.** We evaluate distributed CNN training with 10 clients in two configurations: (1) CIFAR-10 dataset with ResNet-9; (2) CIFAR-100 with ResNet-18 [54, 64]. In both tasks, DRIVE and DRIVE$^+$ have similar accuracy to FedAvg, closely followed by Kashin + 1-bit SQ. The other algorithms are less accurate, with Hadamard + 1-bit SQ being better than Sketched-SGD and TernGrad for both tasks. Additional details and hyperparameter configurations are presented in Appendix E.5. Figure 3 depicts the results.

**Evaluation Summary.** Overall, it is evident that DRIVE and DRIVE$^+$ consistently offer markedly favorable results in comparison to the alternatives in our setting. Kashin's representation appears to offer the best competition, albeit at somewhat higher computational complexity and bandwidth requirements. The lesser performance of the sketch-based techniques is attributed to the high noise of the sketch under such a low ($d(1+o(1))$ bits) communication requirement. This is because the number of counters they can use is too low, making too many coordinates map into each counter. In Appendix E.6, we also compare DRIVE and DRIVE$^+$ to state of the art techniques over K-Means and Power Iteration tasks for 10, 100, and 1000 clients, yielding similar trends.

# 8 Discussion

**Proven Error Bounds.** We summarize the proven error bounds in Table 2. Since DRIVE (Hadamard) is generally not unbiased (as discussed in Section 6.2), we cannot establish a formal guarantee for the 1b - DME problem when using Hadamard. It is a challenging research question whether there exists other structured rotations with low computational complexity and stronger guarantees.

| Problem | Scale $S$ | Rotation | |
|---|---|---|---|
| | | Uniform | Hadamard |
| 1b - VE | $\frac{\|\mathcal{R}(x)\|_1}{d}$ | $\text{vNMSE} = \left(1 - \frac{2}{\pi}\right)\left(1 - \frac{1}{d}\right)$ | $\text{vNMSE} \leqslant \frac{1}{2}$ |
| 1b - DME | $\frac{\|x\|_2^2}{\|\mathcal{R}(x)\|_1}$ | $\text{NMSE} \leqslant \frac{1}{n} \cdot 2.92; \quad d \geqslant 135 \implies \text{NMSE} \leqslant \frac{1}{n} \cdot \left(\frac{\pi}{2} - 1 + \sqrt{\frac{(6\pi^3 - 12\pi^2)\cdot \ln d + 1}{d}}\right)$ | — |

Table 2: Summary of the proven error bounds for DRIVE.

**Input Distribution Assumption.** The distributed mean estimation analysis of our Hadamard-based variants is based on an assumption (Section 6.2) about the vector distributions. While machine learning workloads, and DNN gradients in particular (e.g., [35, 36, 37]), were observed to follow such distributions, this assumption may not hold for other applications.

For such cases, we note that DRIVE is compatible with the error feedback (EF) mechanism [65, 66] that ensured convergence and recovery of the convergence rate of non-compressed SGD. Specifically, as evident by Lemma 3, any scale $\frac{\|\mathcal{R}(x)\|_1}{d} \leqslant S \leqslant 2 \cdot \frac{\|\mathcal{R}(x)\|_1}{d}$ is sufficient to respect the *compressor* (i.e., *bounded variance*) assumption. For completeness, in Appendix E.2, we perform EF experiments comparing DRIVE and DRIVE$^+$ to other compression techniques that use EF.

**Varying Communication Budget.** Unlike some previous works, our algorithms' guarantees with more than one bit per coordinate are not established. It is thus an interesting future work to extend DRIVE to other communication budgets and understand what are the resulting guarantees. We refer the reader to [67] for initial steps towards that direction.

**Entropy Encoding.** Entropy encoding methods (such as Huffman coding) can further compress vectors of values when the values are not uniformly distributed. We have compared DRIVE against stochastic quantization methods using entropy encoding for the challenging setting for DRIVE where all vectors are the same (see Table 1 for further description). The results appear in Appendix E.7, where DRIVE still outperforms these methods. We also note that, when computation allows and when using DRIVE with multiple bits per entry, DRIVE can also be enhanced by entropy encoding techniques. We describe some initial results for this setting in [67].

**Structured Data.** When the data is highly sparse, skewed, or otherwise structured, one can leverage that for compression. We note that some techniques that exploit sparsity or structure can be use in conjunction with our techniques. For example, one may transmit only non-zero entries or Top-K entries while compressing these using DRIVE to reduce communication overhead even further.

**Compatibility With Distributed All-Reduce Techniques.** Quantization techniques, including DRIVE, may introduce overheads in the context of All-Reduce (depending on the network architecture and communication patterns). In particular, if every node in a cluster uses a different rotation, DRIVE will not allow for efficient in-path aggregation without decoding the vectors. Further, the computational overhead of the receiver increases by a $\log d$ factor as each vector has to be decoded separately before an average can be computed. It is an interesting future direction for DRIVE to understand how to minimize such potential overheads. For example, one can consider bucketizing co-located workers and apply DRIVE's quantization only for cross-rack traffic.

## 9   Conclusions

In this paper, we studied the vector and distributed mean estimation problems. These problems are applicable to distributed and federated learning, where clients communicate real-valued vectors (e.g., gradients) to a server for averaging. To the best of our knowledge, our algorithms are the first with a provable error of $O(\frac{1}{n})$ for the 1b - Distributed Mean Estimation problem (i.e., with $d(1 + o(1))$ bits). As shown in [14], any algorithm that uses $O(d)$ shared random bits (e.g., our Hadamard-based variant) has a vNMSE of $\Omega(1)$, i.e., DRIVE and DRIVE$^+$ are asymptotically optimal; additional discussion is given in Appendix F. Our experiments, carried over various tasks and datasets, indicate that our algorithms improve over the state of the art. All the results presented in this paper are fully reproducible by our source code, available at [29].

## Acknowledgments and Disclosure of Funding

MM was supported in part by NSF grants CCF-2101140, CCF-2107078, CCF-1563710, and DMS-2023528. MM and RBB were supported in part by a gift to the Center for Research on Computation and Society at Harvard University. AP was supported in part by the Cyber Security Research Center at Ben-Gurion University of the Negev. We thank Moshe Gabel, Mahmood Sharif, Yuval Filmus and, the anonymous reviewers for helpful comments and suggestions.

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
