*Proof.* Let $T \in \mathcal{S}^{d-1}$ be uniformly distributed on the unit sphere. By the definition of $\mathcal{R}_U$, for *any* $x \in \mathbb{R}^d$, $\mathcal{R}_U(\breve{x})$ and $T$ follow the same distribution (i.e., $\mathcal{R}_U(\breve{x})$ is also uniformly distributed on the unit sphere). As was established in [67], a uniformly distributed point on a sphere $T \in \mathcal{S}^{d-1}$ can be obtained by first deriving a random vector $Z = (Z_1, \ldots, Z_d)$ where $Z_i \sim N(0,1), i \in [1, \ldots, d]$ are normally distributed i.i.d. random variables, and then normalizing its norm by $T = \frac{Z}{\|Z\|_2}$. We also make use of the fact that the norm and direction of a standard multivariate normal vector are independent. That is, $T$ and $\|Z\|_2$ are independent. Therefore, we have that, $\|T\|_1^2 = \left\|\frac{Z}{\|Z\|_2}\right\|_1^2 = \frac{1}{\|Z\|_2^2} \cdot \|Z\|_1^2$ and thus $\|T\|_1^2 \cdot \|Z\|_2^2 = \|Z\|_1^2$. Taking expectation and rearranging yields,

$$\mathbb{E}\left[\|T\|_1^2\right] = \frac{\mathbb{E}\left[\|Z\|_1^2\right]}{\mathbb{E}\left[\|Z\|_2^2\right]} = \frac{\mathbb{E}\left[\sum_{i=1}^d Z_i^2 + \sum_{(i,j) \in \{d \times d\}, i \neq j} |Z_i| \cdot |Z_j|\right]}{d}$$

$$= \frac{\sum_{i=1}^d \mathbb{E}\left[Z_i^2\right] + \sum_{(i,j) \in \{d \times d\}, i \neq j} \mathbb{E}\left[|Z_i|\right] \cdot \mathbb{E}\left[|Z_j|\right]}{d} = \frac{d + d \cdot (d-1) \cdot (\sqrt{\frac{2}{\pi}})^2}{d} = 1 + (d-1)\frac{2}{\pi}.$$

Here we employed $\mathbb{E}\left[Z_i^2\right] = 1$ and that $|Z_i|$ and $|Z_j|$ are independent half-normal random variables for $i \neq j$ with $\mathbb{E}\left[|Z_i|\right] = \mathbb{E}\left[|Z_j|\right] = \sqrt{\frac{2}{\pi}}$. Finally, the resulting vNMSE depends only on the dimension and not the specific vector $x$. In particular, $1 - \mathbb{E}\left[\mathcal{L}_{\mathcal{R},x}^d\right] = 1 - \frac{1+(d-1)\frac{2}{\pi}}{d} = \left(1 - \frac{2}{\pi}\right)\left(1 - \frac{1}{d}\right)$. $\square$

## A.2 Proof of Theorem 3

**Theorem 3.** *For any $x \in \mathbb{R}^d$, set $S = \frac{\|x\|_2^2}{\|\mathcal{R}_U(x)\|_1}$. Then DRIVE satisfies $\mathbb{E}[\hat{x}] = x$.*

*Proof.* For any $x \in \mathbb{R}^d$ denote $x' = (\|x\|_2, 0, \ldots, 0)^T$ and let $R_{x \to x'} \in \mathbb{R}^{d \times d}$ be a rotation matrix such that $R_{x \to x'} \cdot x = x'$. Further, denote $R_x = R_U R_{x \to x'}^{-1}$.

Using these definitions and observing that $\|\mathcal{R}_U(x)\|_1 = \langle R_U \cdot x, \text{sign}(R_U \cdot x)\rangle$ we obtain:

$$\begin{aligned}
\hat{x} &= R_{x \to x'}^{-1} \cdot R_{x \to x'} \cdot \hat{x} = R_{x \to x'}^{-1} \cdot R_{x \to x'} \cdot R_U^{-1} \cdot S \cdot \text{sign}\left(R_U \cdot x\right) \\
&= R_{x \to x'}^{-1} \cdot R_x^{-1} \cdot S \cdot \text{sign}\left(R_x \cdot R_{x \to x'} \cdot x\right) = S \cdot R_{x \to x'}^{-1} \cdot R_x^{-1} \cdot \text{sign}\left(R_x \cdot x'\right) \\
&= \frac{\|x\|_2^2}{\langle R_U \cdot x, \text{sign}(R_U \cdot x)\rangle} \cdot R_{x \to x'}^{-1} \cdot R_x^{-1} \cdot \text{sign}\left(R_x \cdot x'\right) \\
&= R_{x \to x'}^{-1} \cdot \|x\|_2^2 \cdot \frac{R_x^{-1} \cdot \text{sign}\left(R_x \cdot x'\right)}{\langle R_x \cdot x', \text{sign}(R_x \cdot x')\rangle}.
\end{aligned} \tag{4}$$

Next, let $C_i$ be a vector containing the values of the $i$'th column of $R_x$. Then, $R_x \cdot x' = \|x\|_2 \cdot C_0$ and $\text{sign}(R_x \cdot x') = \text{sign}(\|x\|_2 \cdot C_0) = \text{sign}(C_0)$. This means that,

$$\langle R_x \cdot x', \text{sign}(R_x \cdot x')\rangle = \|x\|_2 \cdot \langle C_0, \text{sign}(C_0)\rangle = \|x\|_2 \cdot \|C_0\|_1. \tag{5}$$

Now, since $R_x^{-1} = R_x^T$, we have that,

$$R_x^{-1} \cdot \text{sign}\left(R_x \cdot x'\right) = (\|C_0\|_1, \langle C_1, \text{sign}(C_0)\rangle, \ldots, \langle C_{d-1}, \text{sign}(C_0)\rangle)^T. \tag{6}$$

Using Eq. (5) and (6) in Eq. (4) yields

$$\hat{x} = R_{x \to x'}^{-1} \cdot \|x\|_2 \cdot \left(1, \frac{\langle C_1, \text{sign}(C_0)\rangle}{\|C_0\|_1}, \ldots, \frac{\langle C_{d-1}, \text{sign}(C_0)\rangle}{\|C_0\|_1}\right)^T. \tag{7}$$

Next, by drawing insight from (7), we show that the estimate of DRIVE is unbiased by constructing a symmetric algorithm to DRIVE whose reconstruction's expected value is identical to that of DRIVE, but with the additional property that the average of both reconstructions is exactly $x$ for each specific run of the algorithm.

In particular, consider an algorithm DRIVE$'$ that operates exactly as DRIVE but, instead of directly using the sampled rotation matrix $R_U = R_x \cdot R_{x \to x'}^{-1}$ it calculates and uses the rotation matrix $R_U' = R_x \cdot I' \cdot R_{x \to x'}^{-1}$ where $I'$ is identical to the $d$-dimensional identity matrix with the exception that $I_{00}' = -1$ instead of 1.

Now, since $R_U \sim \mathcal{R}_U$ and $R_{x \to x'}$ is a fixed rotation matrix, we have that $R_x = R_U \cdot R_{x \to x'} \sim \mathcal{R}_U$. In turn, this also means that $R_x \cdot I' \sim \mathcal{R}_U$ since $I'$ is a fixed rotation matrix.

Consider a run of both algorithm where $\widehat{x}$ is the reconstruction of DRIVE for $x$ with a sampled rotation $R_U$ and $\widehat{x}'$ is the corresponding reconstruction of DRIVE$'$ for $x$ with the rotation $R_U'$.

According to (7) it holds that: $\widehat{x} + \widehat{x}' = R_{x \to x'}^{-1} \cdot \|x\|_2 \cdot (2, 0, \ldots, 0)^T = 2 \cdot x$. This is because both runs are identical except that the first column of $R_x$ (i.e., $C_0$) and $R_x \cdot I'$ have opposite signs. In particular, it holds that $\mathbb{E}\left[\widehat{x} + \widehat{x}'\right] = 2 \cdot x$. But, since $R_x \sim \mathcal{R}_U$ and $R_x \cdot I' \sim \mathcal{R}_U$, both algorithms have the same expected value. This yields $\mathbb{E}\left[\widehat{x}\right] = \mathbb{E}\left[\widehat{x}'\right] = x$. This concludes the proof. $\qquad\square$

### A.3 Proof of Theorem 4

The proof of the theorem follows from the following two lemmas. Lemma 6 relies on Chebyshev's inequality and allows us to bound the vNMSE for all $d$. Lemma 7 gives a sharper bound for large dimensions using the Bernstein's inequality.

**Lemma 6.** *For any constant $k$ and dimension $d \geqslant 2$ such that $0 < k < \frac{d}{d-1} \cdot \sqrt{\frac{\pi}{\pi-3}} \cdot \frac{2}{\mathrm{B}(\frac{1}{2}, \frac{d-1}{2})}$,*

$$\mathbb{E}\left[\frac{d}{\|\mathcal{R}(\check{x})\|_1^2}\right] \leqslant \frac{d}{k^2} + \frac{\left(1 - \frac{1}{k^2}\right)}{\left(-k \cdot \sqrt{\frac{\pi-3}{\pi d}} + \frac{2 \cdot \sqrt{d}}{(d-1) \cdot \mathrm{B}(\frac{1}{2}, \frac{d-1}{2})}\right)^2} \ .$$

*Proof.* By Lemma 9 in Appendix A.4 we have that $\mathbb{E}\left[\|R_U \cdot \check{x}\|_1\right] = \frac{2d}{(d-1) \cdot \mathrm{B}(\frac{1}{2}, \frac{d-1}{2})}$. Thus,

$$
\begin{aligned}
\mathrm{Var}\left(\frac{\|R_U \cdot \check{x}\|_1}{\sqrt{d}}\right) &= \mathbb{E}\left[\frac{\|R_U \cdot \check{x}\|_1^2}{d}\right] - \left(\mathbb{E}\left[\frac{\|R_U \cdot \check{x}\|_1}{\sqrt{d}}\right]\right)^2 \\
&= \frac{2}{\pi} + \frac{1 - \frac{2}{\pi}}{d} - \left(\frac{2 \cdot \sqrt{d}}{(d-1) \cdot \mathrm{B}(\frac{1}{2}, \frac{d-1}{2})}\right)^2 \leqslant \frac{\pi-3}{\pi d} .
\end{aligned}
$$

According to Chebyshev's inequality, $\Pr\left[-\frac{\|R_U \cdot \check{x}\|_1}{\sqrt{d}} + \frac{2 \cdot \sqrt{d}}{(d-1) \cdot \mathrm{B}(\frac{1}{2}, \frac{d-1}{2})} \geqslant k \cdot \sqrt{\frac{\pi-3}{\pi d}}\right] \leqslant \frac{1}{k^2}$ which is equivalent to $\Pr\left[\frac{\|R_U \cdot \check{x}\|_1}{\sqrt{d}} \leqslant -k \cdot \sqrt{\frac{\pi-3}{\pi d}} + \frac{2 \cdot \sqrt{d}}{(d-1) \cdot \mathrm{B}(\frac{1}{2}, \frac{d-1}{2})}\right] \leqslant \frac{1}{k^2}$. Now, for any $k$ that respects $-k \cdot \sqrt{\frac{\pi-3}{\pi d}} + \frac{2 \cdot \sqrt{d}}{(d-1) \cdot \mathrm{B}(\frac{1}{2}, \frac{d-1}{2})} > 0$ it holds that $\Pr\left[\frac{d}{\|R_U \cdot \check{x}\|_1^2} \geqslant \frac{1}{\left(-k \cdot \sqrt{\frac{\pi-3}{\pi d}} + \frac{2 \cdot \sqrt{d}}{(d-1) \cdot \mathrm{B}(\frac{1}{2}, \frac{d-1}{2})}\right)^2}\right] \leqslant \frac{1}{k^2}$.

In addition, observe that $\Pr\left[\frac{d}{\|R_U \cdot \check{x}\|_1^2} > d\right] = 0$. This means that,

$$\mathbb{E}\left[\frac{d}{\|R_U \cdot \check{x}\|_1^2}\right] \leqslant \frac{d}{k^2} + \left(1 - \frac{1}{k^2}\right) \cdot \left(-k \cdot \sqrt{\frac{\pi-3}{\pi d}} + \frac{2 \cdot \sqrt{d}}{(d-1) \cdot \mathrm{B}(\frac{1}{2}, \frac{d-1}{2})}\right)^{-2} \ .$$

This concludes the proof. $\qquad\square$

**Lemma 7.** *For any vector $x \in \mathbb{R}^d$, a constant $0 < \epsilon < 1$, and any dimension $d \geqslant 2$, it holds that*

$$\mathbb{E}\left[\frac{d}{\|\mathcal{R}(\check{x})\|_1^2}\right] \leqslant \frac{\pi}{2} \cdot (1 + \epsilon) + d \cdot exp\left(-d \cdot \frac{\epsilon^2}{16(\pi-2)}\right) \ .$$

*Proof.* By the definition of a uniform random rotation, it holds that $\frac{d}{\|\mathcal{R}(\check{x})\|_1^2}$ follows the same distribution as $\frac{d \cdot \|Z\|_2^2}{\|Z\|_1^2}$ where $Z = (Z_1, \ldots, Z_d)$ and $Z_i \sim \mathcal{N}(0, 1)$ are i.i.d. normal variables. Consider the two complementary events, $A_- = \|Z\|_1 \leqslant \beta \cdot d$ and $A_+ = \|Z\|_1 > \beta \cdot d$. Then,

$$\mathbb{E}\left[\frac{d \cdot \|Z\|_2^2}{\|Z\|_1^2}\right] = \mathbb{E}\left[\mathbb{1}_{A_+} \cdot \frac{d \cdot \|Z\|_2^2}{\|Z\|_1^2}\right] + \mathbb{E}\left[\mathbb{1}_{A_-} \cdot \frac{d \cdot \|Z\|_2^2}{\|Z\|_1^2}\right].$$

We now treat each of the terms separately. For the first term we have,

$$\mathbb{E}\left[\mathbb{1}_{A_+} \cdot \frac{d \cdot \|Z\|_2^2}{\|Z\|_1^2}\right] \leqslant \frac{d}{(d \cdot \beta)^2} \cdot \mathbb{E}\left[\|Z\|_2^2\right] = \frac{1}{\beta^2}.$$

For the second term we have,

$$\mathbb{E}\left[\mathbb{1}_{A_-} \cdot \frac{d \cdot \|Z\|_2^2}{\|Z\|_1^2}\right] \leqslant d \cdot \mathbb{E}\left[\mathbb{1}_{A_-}\right] = d \cdot \Pr\left[\|Z\|_1 \leqslant \beta \cdot d\right].$$

Now, our goal is to bound the term $\Pr\left[\|Z\|_1 \leqslant \beta \cdot d\right]$. Denote $Y = (Y_1, \ldots, Y_d)$, where $Y_i = \sqrt{\frac{2}{\pi}} - |Z_i|$. Then, it holds that,

$$\Pr\left[\|Z\|_1 \leqslant \beta d\right] = \Pr\left[\|Z\|_1 - d\sqrt{\frac{2}{\pi}} \leqslant \beta d - d\sqrt{\frac{2}{\pi}}\right] = \Pr\left[\sum_{i=1}^d Y_i \geqslant d\left(\sqrt{\frac{2}{\pi}} - \beta\right)\right].$$

Further, for all $i$ we have that $\mathbb{E}[Y_i] = 0$, $\mathbb{E}[Y_i]^2 = 1 - \frac{2}{\pi}$ and $Y_i \leqslant \sqrt{\frac{2}{\pi}}$. According to Bernstein's Inequality (from [68], or see [69, Theorem 3.1] for a simplified notation) this means that,

$$\Pr\left[\sum_{i=1}^d Y_i \geqslant d\left(\sqrt{\frac{2}{\pi}} - \beta\right)\right] \leqslant exp\left(\frac{-d^2 \cdot (\sqrt{\frac{2}{\pi}} - \beta)^2}{2d \cdot (1 - \frac{2}{\pi}) + d \cdot \frac{2}{3} \cdot \sqrt{\frac{2}{\pi}} \cdot \left(\sqrt{\frac{2}{\pi}} - \beta\right)}\right).$$

Setting $\beta = \sqrt{\frac{2-\epsilon}{\pi}}$ for some $\epsilon > 0$ yields,

$$\Pr\left[\sum_{i=1}^d Y_i \geqslant d\left(\sqrt{\frac{2}{\pi}} - \beta\right)\right] \leqslant exp\left(-d \cdot \frac{3 \cdot \left(1 - \sqrt{1 - \frac{\epsilon}{2}}\right)^2}{3\pi - 4 - 2\sqrt{1 - \frac{\epsilon}{2}}}\right).$$

Next, we obtain,

$$\mathbb{E}\left[\frac{d \cdot \|Z\|_2^2}{\|Z\|_1^2}\right] \leqslant \frac{\pi}{2} \cdot \frac{1}{1 - \frac{\epsilon}{2}} + d \cdot exp\left(-d \cdot \frac{3 \cdot \left(1 - \sqrt{1 - \frac{\epsilon}{2}}\right)^2}{3\pi - 4 - 2\sqrt{1 - \frac{\epsilon}{2}}}\right).$$

We next use Taylor expansions to simplify the expression:

$$\mathbb{E}\left[\frac{d \cdot \|Z\|_2^2}{\|Z\|_1^2}\right] \leqslant \frac{\pi}{2} \cdot \frac{1}{1 - \frac{\epsilon}{2}} + d \cdot exp\left(-d \cdot \frac{3 \cdot \left(1 - \sqrt{1 - \frac{\epsilon}{2}}\right)^2}{3\pi - 4 - 2\sqrt{1 - \frac{\epsilon}{2}}}\right)$$

$$\leqslant \frac{\pi}{2} \cdot \frac{1}{1 - \frac{\epsilon}{2}} + d \cdot exp\left(-d \cdot \frac{\left(\frac{\epsilon}{2}\right)^2}{4(\pi - 2)}\right)$$

$$\leqslant \frac{\pi}{2} \cdot \left(1 + 2 \cdot \frac{\epsilon}{2}\right) + d \cdot exp\left(-d \cdot \frac{\left(\frac{\epsilon}{2}\right)^2}{4(\pi - 2)}\right)$$

$$= \frac{\pi}{2} \cdot (1 + \epsilon) + d \cdot exp\left(-d \cdot \frac{\epsilon^2}{16(\pi - 2)}\right).$$

This concludes the proof. $\qquad\square$

Finally, we prove the theorem, which we restate here.

**Theorem 4.** *For any $x \in \mathbb{R}^d$, the vNMSE of DRIVE with $S = \frac{\|x\|_2^2}{\|\mathcal{R}_U(x)\|_1}$ satisfies:*

*(i) For all $d \geqslant 2$, it is at most 2.92. (ii) For all $d \geqslant 135$, it is at most $\frac{\pi}{2} - 1 + \sqrt{\frac{(6\pi^3 - 12\pi^2) \cdot \ln d + 1}{d}}$.*

*Proof.* Recall that, using Lemma 2, the vNMSE is $\mathbb{E}\left[\frac{d}{\|\mathcal{R}(\check{x})\|_1^2}\right] - 1$. Therefore, the first part of the theorem follows from Lemma 6 with $k = \sqrt{d}$, which satisfies $\sqrt{k} < \frac{d-1}{d} \cdot \sqrt{\frac{\pi}{\pi-3}} \cdot \frac{2}{B(\frac{1}{2}, \frac{d-1}{2})}$, as needed. The second part of the theorem follows from setting $\epsilon = \sqrt{\frac{(24\pi - 48)\ln d}{d}}$ in Lemma 7. Notice that for $d \geqslant 135$, we get $\epsilon < 1$, as required. Then:

$$\mathbb{E}\left[\frac{d}{\|\mathcal{R}(\check{x})\|_1^2}\right] \leqslant \frac{\pi}{2} \cdot (1 + \epsilon) + d \cdot exp\left(-1.5 \ln d\right) = \frac{\pi}{2} + \sqrt{\frac{(6\pi^3 - 12\pi^2) \cdot \ln d + 1}{d}} .$$

This concludes the proof. $\qquad\square$

## A.4 Constant Scale for Unbiased Estimates in DRIVE with Uniform Random Rotations

In this appendix, we prove that it is possible to further lower the vNMSE while remaining unbiased by (deterministically) using $S = \frac{\|x\|_2^2}{\mathbb{E}\left[\|\mathcal{R}_U(x)\|_1\right]} = \frac{\|x\|_2}{\mathbb{E}\left[\|T\|_1\right]}$ where $T \in \mathcal{S}^{d-1}$ is uniformly at random distributed on the unit sphere (observe that the unbiasedness proof in Theorem A.2 holds for this constant scale). This is because $\frac{d}{(\mathbb{E}\left[\|T\|_1\right])^2} - 1 \leqslant \mathbb{E}\left[\frac{d}{\|T\|_1^2}\right] - 1$. This gives a provable $\frac{\pi}{2} - 1$ vNMSE bound for any $d$. We start the analysis by proving two auxiliary lemmas. A generalization of these results is proved in [70, Eq. 3.3]. For completeness, we provide simplified proofs.

**Lemma 8.** *Let $T = (T_1, \ldots, T_d) \in \mathcal{S}^{d-1}$ be a point on the unit sphere, drawn uniformly at random. Then, the PDF of $|T_i|$ is given by,*

$$f_{|T_i|}(t) = \frac{2 \cdot (1 - t^2)^{\frac{d-1}{2} - 1}}{B(\frac{1}{2}, \frac{d-1}{2})} .$$

*Proof.* As was established in [67], a uniformly random point on a sphere $T$ can be obtained by first deriving a random vector $Z = (Z_1, \ldots, Z_d)$ where $Z_i \sim N(0, 1), i \in [1, \ldots, d]$ are normally distributed i.i.d. random variables, and then normalizing its norm by $T = \frac{Z}{\|Z\|_2}$.

Therefore, the i'th coordinate of $T$ is given as $T_i = \frac{Z_i}{\|Z\|_2}$. Since the coordinate distribution is sign-symmetric, we focus on its distribution in the interval $[0, 1]$. Observe that for $T_i \in [0, 1]$ :

$$\Pr\left[|T_i| \leqslant t\right] = \Pr\left[T_i^2 \leqslant t^2\right] = \Pr\left[\frac{Z_i^2}{\|Z\|_2^2} \leqslant t^2\right] = \Pr\left[\frac{Z_i^2}{\|Z\|_2^2 - Z_i^2} \leqslant t^2 \cdot \frac{\|Z\|_2^2}{\|Z\|_2^2 - Z_i^2}\right]$$

$$= \Pr\left[\frac{Z_i^2}{\|Z\|_2^2 - Z_i^2} \leqslant t^2 \cdot \left(1 + \frac{Z_i^2}{\|Z\|_2^2 - Z_i^2}\right)\right] = \Pr\left[\frac{Z_i^2}{\|Z\|_2^2 - Z_i^2} \leqslant \frac{t^2}{1 - t^2}\right] .$$

Therefore $\Pr\left[|T_i| \leqslant t\right] = \Pr\left[\frac{(d-1)Z_i^2}{\|Z\|_2^2 - Z_i^2} \leqslant \frac{(d-1)t^2}{1 - t^2}\right]$. Notice that $Z_i^2 \sim \chi^2(1)$ and $(\|Z\|_2^2 - Z_i^2) \sim \chi^2(d-1)$ are independent $\chi^2$ random variables. Now denote $Y \triangleq \frac{(d-1)Z_i^2}{\|Z\|_2^2 - Z_i^2}$. Observe that $Y$ follows a $F_{1, d-1}$ distribution. Therefore, the CDF of $Y$ is then given by the following expression: $\Pr\left[Y \leqslant y\right] = I_{\frac{y}{y+d-1}}(\frac{1}{2}, \frac{d-1}{2})$, where $I$ is the regularized incomplete Beta function. Substituting $y = \frac{(d-1)t^2}{1-t^2}$ yields $\Pr\left[|T_i| \leqslant t\right] = I_{t^2}(\frac{1}{2}, \frac{d-1}{2})$. In particular, taking the derivative yields $f_{|T_i|}(t) = \frac{2 \cdot (1-t^2)^{\frac{d-1}{2}-1}}{B(\frac{1}{2}, \frac{d-1}{2})}$. Interestingly, this PDF is similar to the Beta distribution (not to be confused with the B function). Indeed, one can verify that $T_i' = \frac{T_i + 1}{2}$ follows a Beta distribution with $T_i' \sim Beta(\frac{d-1}{2}, \frac{d-1}{2})$. $\quad\square$

With Lemma 8 at hand, we obtain the following result.

**Lemma 9.** *Let $T \in \mathcal{S}^{d-1}$ be a point on the unit sphere, drawn uniformly at random. Then,*

$$\mathbb{E}\left[\|T\|_1\right] = \frac{2d}{(d-1) \cdot \mathrm{B}(\frac{1}{2}, \frac{d-1}{2})} \ .$$

*Proof.* Due to the linearity of expectation we have that $\mathbb{E}\left[\|T\|_1\right] = \sum_{i=1}^{d} \mathbb{E}\left[|T_i|\right]$. Thus, it is sufficient to calculate the expected value of a single element in the sum. By Lemma 8 we have that $\mathbb{E}\left[|T_i|\right] = \int_0^1 t \cdot \frac{2 \cdot (1-t^2)^{\frac{d-1}{2}-1}}{\mathrm{B}(\frac{1}{2}, \frac{d-1}{2})} dt = \frac{2}{(d-1) \cdot \mathrm{B}(\frac{1}{2}, \frac{d-1}{2})}$. Summing over $i \in \{1, \ldots, d\}]$ yields the result. $\square$

We note that $\mathbb{E}\left[\|T\|_1\right] = \frac{2d}{(d-1) \cdot \mathrm{B}(\frac{1}{2}, \frac{d-1}{2})} \geqslant \sqrt{\frac{2d}{\pi}}$. Finally, recall that our vNMSE is bounded by $\frac{d}{\mathbb{E}\left[\|T\|_1^2\right]} - 1$, which is therefore at most $\frac{\pi}{2} - 1$. In practice, while this deterministic approach gives a lower vNMSE for low dimensions, the benefit is marginal even for $d$ values of several hundreds.

### A.5 Proof of Theorem 5

**Theorem 5.** *Assume $n$ clients, each with its own vector $x_{(c)} \in \mathbb{R}^d$. Let each client independently sample $R_{U,c} \sim \mathcal{R}_U$ and set its scale to $\frac{\|x_{(c)}\|_2^2}{\|R_{U,c} \cdot x_{(c)}\|_1}$. Then, the server average estimate's NMSE satisfies: $\frac{\mathbb{E}\left[\|x_{avg} - \widehat{x_{avg}}\|_2^2\right]}{\frac{1}{n} \cdot \sum_{c=1}^{n} \|x_{(c)}\|_2^2} = \frac{vNMSE}{n}$, where vNMSE is given by Lemma 2 and is bounded by Theorem 4.*

*Proof.* We have that,

$$\mathbb{E}\left[\|x_{avg} - \widehat{x_{avg}}\|_2^2\right] = \frac{1}{n^2} \cdot \mathbb{E}\left[\left\|\sum_{c=1}^{n} x_{(c)} - \sum_{c=1}^{n} \widehat{x_{(c)}}\right\|_2^2\right] = \frac{1}{n^2} \cdot \sum_{c,c'} \mathbb{E}\left[\left\langle x_{(c)} - \widehat{x_{(c)}}, x_{(c')} - \widehat{x_{(c')}}\right\rangle\right]$$

$$= \frac{1}{n^2} \cdot \sum_c \mathbb{E}\left[\left\langle x_{(c)} - \widehat{x_{(c)}}, x_{(c)} - \widehat{x_{(c)}}\right\rangle\right]$$

$$+ \frac{1}{n^2} \cdot \sum_{c \neq c'} \mathbb{E}\left[\left\langle x_{(c)} - \widehat{x_{(c)}}, x_{(c')} - \widehat{x_{(c')}}\right\rangle\right]$$

$$\overset{\langle 1 \rangle}{=} \frac{1}{n^2} \sum_c \mathbb{E}\left[\|x_{(c)} - \widehat{x_{(c)}}\|_2^2\right] = \frac{1}{n^2} \cdot \sum_c \|x_{(c)}\|_2^2 \cdot \mathbb{E}\left[\frac{\|x_{(c)} - \widehat{x_{(c)}}\|_2^2}{\|x_{(c)}\|_2^2}\right]$$

$$= \frac{1}{n^2} \cdot \sum_c \|x_{(c)}\|_2^2 \cdot vNMSE = \frac{vNMSE}{n} \cdot \frac{\sum_c \|x_{(c)}\|_2^2}{n}.$$

Here, since the $\mathbb{E}\left[\widehat{x_{(c)}}\right] = x_{(c)}$ for all $i$ and since $\widehat{x_{(c)}}$ and $\widehat{x_{(c')}}$ are independent for all $c \neq c'$, $\langle 1 \rangle$ follows from $\mathbb{E}\left[\left\langle x_{(c)} - \widehat{x_{(c)}}, x_{(c')} - \widehat{x_{(c')}}\right\rangle\right] = \mathbb{E}\left[\left\langle x_{(c)}, x_{(c')}\right\rangle\right] - \mathbb{E}\left[\left\langle x_{(c)}, \widehat{x_{(c')}}\right\rangle\right] - \mathbb{E}\left[\left\langle \widehat{x_{(c)}}, x_{(c')}\right\rangle\right] + \mathbb{E}\left[\left\langle \widehat{x_{(c)}}, \widehat{x_{(c')}}\right\rangle\right] = 0$. $\square$

With this result at hand, we can derive useful theoretical guarantees, especially for high-dimensional vectors (e.g., gradient updates in NN federated and distributed training). For example

**Corollary 2.** *For $n$ clients with arbitrary vectors in $\mathbb{R}^d$ and $d \geqslant 10^5$, the NMSE is smaller than $\frac{0.673}{n}$.*

## B Message Representation Length

Here we discuss the length of the message $\left(S, \mathrm{sign}(\mathcal{R}(x))\right)$ that Buffy sends to Angel. The message includes the sign vector $\mathrm{sign}(\mathcal{R}(x))$, which takes exactly $d$ bits. It remains to consider how $S$ is to be represented and how that affects the vNMSE of our algorithms. We first note that one must assume that the norm of the sent vector $x$ is bounded, as otherwise no finite number of bits can be used for the transmission. This is justified as the coordinates of the vector are commonly represented by a

floating point representation with constant length (e.g., using 32 bits). In particular, this assumption implies that $\|x\|_2 = O(d)$, with each coordinate is bounded by a constant. It follows that all the quantities we consider for the scale satisfy $S = O(d)$. This is because the scale of Theorem 2 satisfies $\frac{\|\mathcal{R}(x)\|_1}{d} \leqslant \frac{\sqrt{d} \cdot \|\mathcal{R}(x)\|_2}{d} = \frac{\|x\|_2}{\sqrt{d}} = O(\sqrt{d})$ (where here the first step uses the Cauchy-Schwarz inequality), and similarly the scale of Theorem 3 satisfies $\frac{\|x\|_2^2}{\|\mathcal{R}(x)\|_1} \leqslant \frac{\|x\|_2^2}{\|\mathcal{R}(x)\|_2} = \|x\|_2 = O(d)$.

As shown in [2], this means that one can use $O(\log d)$ bits to represent $S$ to within a $d^{-\Omega(1)}$ additive error, i.e., a factor that rapidly drops to zero as $d$ grows. This increases the vNMSE for a single vector under DRIVE by at most $d^{-\Omega(1)}$, i.e., an additive factor that diminishes as $d$ grows.

In the distributed mean estimation setting, where $n$ clients send their vectors for averaging, [2] suggested using $O(\log(nd))$ bits. In fact, the dependency on $n$ may be dropped. Specifically, $O(\log d)$ bits suffice if we encode $S$ *in an unbiased manner*. One simple way to do so is to encode $\lfloor S \rfloor$ exactly (which requires $O(\log d)$ bits since $S = O(d)$) and to encode the remainder $r = S - \lfloor S \rfloor$ using $O(\log d)$-bit stochastic quantization. (That is, express $r$ using the $O(\log d)$ bits but use randomized rounding for the last bit.) This approach still yields a vNMSE increase of $d^{-\Omega(1)}$ for each vector. As for the distributed mean estimation task, it follows that Theorem 5 still holds for DRIVE as the vectors are sent in an unbiased manner. Thus the NMSE increases by only $(nd)^{-\Omega(1)}$.

The above argument means that DRIVE can use $d + O(\log d) = d(1 + o(1))$ bit messages. While in practice implementations use 32- or 64-bit representations for $S$ thus send $d + O(1)$ bit messages, we use the $d(1 + o(1))$ notation to ensure compatibility with the theoretical guarantees.

Similar arguments show that our improved DRIVE$^+$ algorithm, presented in Section 5, can also be encoded using $d + O(\log d) = d(1 + o(1))$ bits.

# C   Supplementary Material for DRIVE$^+$

In this section, we provide additional information about the DRIVE$^+$ algorithm.

## C.1   Pseudocode

Here, we provide the pseudocode for DRIVE$^+$ in Algorithm 2:

---
**Algorithm 2** DRIVE$^+$

---

**Buffy:**

1: Compute $\mathcal{R}(x)$.
2: Compute the $c_0, c_1 \in \mathbb{R}$ values that minimize

$$\sum_{i=1}^{d} \min \left\{ (\mathcal{R}(x)_i - c_0)^2, (\mathcal{R}(x)_i - c_1)^2 \right\}.$$

3: Compute $X \in \{0, 1\}^d$ such that

$$X_i = \begin{cases} 0 & \text{if } |\mathcal{R}(x)_i - c_0| \leqslant |\mathcal{R}(x)_i - c_1| \\ 1 & \text{otherwise} \end{cases}.$$

4: Compute $S^+$ and $\bar{c}_0 = S^+ \cdot c_0$, $\bar{c}_1 = S^+ \cdot c_1$.
5: Send $(\bar{c}_0, \bar{c}_1, X)$ to Angel.

**Angel:**

1: Compute $\widehat{\mathcal{R}(x)} \in \{\bar{c}_0, \bar{c}_1\}^d$ such that

$$\widehat{\mathcal{R}(x)}_i = \bar{c}_{X_i}.$$

2: Estimate $\hat{x} = \mathcal{R}^{-1}(\widehat{\mathcal{R}(x)})$.

---

We now provide the vNMSE guarantees for DRIVE$^+$.

## C.2   1b - Vector Estimation with DRIVE$^+$

We provide the following lemma from which it follows that the vNMSE of DRIVE$^+$ with $S^+ = 1$ is upper-bounded by that of DRIVE when both algorithms are not required to be unbiased.

**Lemma 10.** *For any $R \sim \mathcal{R}$ and any $x \in \mathbb{R}^d$, the SSE of DRIVE$^+$ with $S^+ = 1$ is upper-bounded by the SSE of DRIVE for any choice of $S$.*

*Proof.* By definition, the values $c_0, c_1$ respect that $\sum_{i=1}^d \min\left\{(\mathcal{R}(x)_i - c_0)^2, (\mathcal{R}(x)_i - c_1)^2\right\} \leqslant \sum_{i=1}^d \min\left\{(\mathcal{R}(x)_i - S)^2, (\mathcal{R}(x)_i + S)^2\right\}$ for any choice of $S$. Now, recall that the SSE in estimating $\mathcal{R}(x)$ using $\widehat{\mathcal{R}(x)}$ equals that of estimating $x$ using $\widehat{x}$. This concludes the proof. $\qquad\square$

## C.3  1b - Distributed Mean Estimation with DRIVE$^+$

In this subsection, we prove that DRIVE$^+$ can be made unbiased, when using the uniform random rotation $\mathcal{R}_U$, by the right choice of scale $S^+$, and that its vNMSE is upper bounded by that of DRIVE.

Let $c \in \{c_0, c_1\}^d$ where $c_0, c_1 \in \mathbb{R}$ minimize the term $\sum_{i=1}^d (\mathcal{R}(x)_i - c_i)^2$. Namely, $c_0, c_1$ are the solution to the 2-means optimization over the entries of the rotated vector.

**Theorem 6.** *For any $x \in \mathbb{R}^d$ set $S^+ = \frac{\|x\|_2^2}{\|c\|_2^2}$. Then, $\mathbb{E}[\widehat{x}] = x$. Thus, Theorem 5 applies to DRIVE$^+$.*

*Proof.* For ease of exposition, for any vector $y \in \mathbb{R}^d$, we denote by $c(y)$ the vector that minimizes the term $\sum_{i=1}^d (y_i - c_i)^2$ where $c(y) \in \{c_0, c_1\}^d$ and $c_0, c_1 \in \mathbb{R}$.

For the vector $\mathcal{R}(x)$ we drop the notation and simply use $c$. Namely, $c \in \{c_0, c_1\}^d$ where $c_0, c_1 \in \mathbb{R}$ is a vector that minimizes the term $\sum_{i=1}^d (\mathcal{R}(x)_i - c_i)^2$.

The proof follows similar lines to that of Theorem 3.

For any $x \in \mathbb{R}^d$ denote $x' = (\|x\|_2, 0, \ldots, 0)^T$ and let $R_{x \to x'} \in \mathbb{R}^{d \times d}$ be a rotation matrix such that $R_{x \to x'} \cdot x = x'$. Further, denote $R_x = R_U R_{x \to x'}^{-1}$. Using these definitions we have that,

$$
\begin{aligned}
\widehat{x} &= R_{x \to x'}^{-1} \cdot R_{x \to x'} \cdot \widehat{x} = R_{x \to x'}^{-1} \cdot R_{x \to x'} \cdot R_U^{-1} \cdot S^+ \cdot c(R_U \cdot x) \\
&= R_{x \to x'}^{-1} \cdot R_x^{-1} \cdot S^+ \cdot c(R_x \cdot R_{x \to x'} \cdot x) = S^+ \cdot R_{x \to x'}^{-1} \cdot R_x^{-1} \cdot c(R_x \cdot x') \\
&= R_{x \to x'}^{-1} \cdot \frac{\|x\|_2^2 \cdot R_x^{-1} \cdot c(R_x \cdot x')}{\|c\|_2^2} .
\end{aligned}
$$

Again, let $C_i$ be a vector containing the values of the $i$'th column of $R_x$. Then, $R_x \cdot x' = \|x\|_2 \cdot C_0$ and therefore $c(R_x \cdot x') = c(\|x\|_2 \cdot C_0) = \|x\|_2 \cdot c(C_0)$. We obtain,

$$
R_x^{-1} \cdot c(R_x \cdot x') = \|x\|_2 \cdot \left(\|c(C_0)\|_2^2, \langle C_1, c(C_0)\rangle, \ldots, \langle C_{d-1}, c(C_0)\rangle\right)^T ,
$$

Next, we have that,

$$
\widehat{x} = R_{x \to x'}^{-1} \cdot \|x\|_2 \cdot \left(\frac{\|x\|_2^2 \cdot \|c(C_0)\|_2^2}{\|c\|_2^2}, \frac{\|x\|_2^2 \cdot \langle C_1, c(C_0)\rangle}{\|c\|_2^2}, \ldots, \frac{\|x\|_2^2 \cdot \langle C_{d-1}, c(C_0)\rangle}{\|c\|_2^2}\right)^T . \tag{8}
$$

Observe that $\|x\|_2^2 \cdot \|c(C_0)\|_2^2 = \|c(R_x \cdot x')\|_2^2 = \|c(R_U \cdot x)\|_2^2 = \|c\|_2^2$. This means that the first coordinate in the above vector is 1.

We continue the proof by using the same construction as in the proof of Theorem 3.

In particular, consider an algorithm DRIVE$'$ that operates exactly as DRIVE but, instead of directly using the sampled rotation matrix $R_U = R_x \cdot R_{x \to x'}^{-1}$ it calculates and uses the rotation matrix $R_U' = R_x \cdot I' \cdot R_{x \to x'}^{-1}$ where $I'$ is identical to the $d$-dimensional identity matrix with the exception that $I_{00}' = -1$ instead of 1.

Now, since $R_U \sim \mathcal{R}_U$ and $R_{x \to x'}$ is a fixed rotation matrix, we have that $R_x \sim \mathcal{R}_U$. In turn, this also means that $R_x \cdot I' \sim \mathcal{R}_U$ since $I'$ is a fixed rotation matrix.

Consider a run of both algorithm where $\widehat{x}$ is the reconstruction of DRIVE for $x$ with a sampled rotation $R_U$ and $\widehat{x}'$ is the corresponding reconstruction of DRIVE$'$ for $x$ with the rotation $R_U'$.

According to (8) it holds that: $\widehat{x} + \widehat{x}' = R_{x \to x'}^{-1} \cdot \|x\|_2 \cdot (2, 0, \ldots, 0)^T = 2 \cdot x$. This is because both runs are identical except that the first column of $R_x$ and $R_x \cdot I'$ have opposite signs and thus the resulting centroids $c(C_0)$ also flip signs in the run of DRIVE'. In particular, it holds that $\mathbb{E}[\widehat{x} + \widehat{x}'] = 2 \cdot x$. But, since $R_x \sim \mathcal{R}_U$ and $R_x \cdot I' \sim \mathcal{R}_U$, both algorithms have the same expected value. This yields $\mathbb{E}[\widehat{x}] = \mathbb{E}[\widehat{x}'] = x$. This concludes the proof. $\qquad\square$

To prove the SSE of DRIVE$^+$ is lower-bounded by that of DRIVE, we use the following observation.

**Observation 3.** *For $i \in \{0, 1\}$, let $I_i = \{j \in \{1, \ldots, d\} \mid |\mathcal{R}(x)_j - c_i| \leqslant |\mathcal{R}(x)_j - c_{1-i}|\}$ denote the set of points closer to $c_i$. Then $c_i = \frac{\sum_{j \in I_i} \mathcal{R}(x)_j}{|I_i|}$.*

We are now ready to prove the bound on the vNMSE DRIVE$^+$.

**Lemma 11.** *For any $d$ and $R_U$, the SSE of DRIVE$^+$ with $S^+ = \frac{\|x\|_2^2}{\|c\|_2^2}$ is upper-bounded by the SSE of DRIVE with $S = \frac{\|x\|_2^2}{\|\mathcal{R}(x)\|_1}$. Thus, Theorem 4 and Corollary 1 apply to DRIVE$^+$.*

*Proof.* Recall that $c \in \{c_0, c_1\}^d$, where $c_0, c_1 \in \mathbb{R}$, minimizes the term $\sum_{i=1}^d (\mathcal{R}(x)_i - c_i)^2$ and that the quantization SSE of the rotated vector equals the SSE of estimating $x$ using $\widehat{x}$. For DRIVE$^+$, this means that the SSE is given by

$$
\begin{aligned}
\left\| \mathcal{R}(x) - S^+ \cdot c \right\|_2^2 &= \|\mathcal{R}(x)\|_2^2 - 2 \cdot S^+ \cdot \langle \mathcal{R}(x), c \rangle + \left\| S^+ \cdot c \right\|_2^2 \\
&= \|x\|_2^2 - 2 \cdot S^+ \cdot \langle \mathcal{R}(x), c \rangle + (S^+)^2 \cdot \|c\|_2^2 \\
&= \|x\|_2^2 - 2 \cdot S^+ \cdot \|c\|_2^2 + (S^+)^2 \cdot \|c\|_2^2 .
\end{aligned}
$$

According to Observation 3, the inner produce in the first coordinate satisfies:

$$
\langle \mathcal{R}(x), c \rangle = \sum_{j \in I_0} \mathcal{R}(x)_j \cdot c_0 + \sum_{j \in I_1} \mathcal{R}(x)_j \cdot c_1 = c_0^2 \cdot |I_0| + c_1^2 \cdot |I_1| = \|c\|_2^2 .
$$

We we will show that for any vector $x \in \mathbb{R}^d$ and any $R_U$, the SSE of DRIVE$^+$ with $S^+ = \frac{\|x\|_2^2}{\|c\|_2^2}$ is upper bounded by the SSE of DRIVE with $S = \frac{\|x\|_2^2}{\|\mathcal{R}_U(x)\|_1}$. Both results immediately follow.

From the above and Theorem 1, it is sufficient to show that $\|c\|_2^2 \left[ (S^+)^2 - 2S^+ \right] \leqslant -2 \cdot S \cdot \|\mathcal{R}(x)\|_1 + S^2 \cdot d$, which, by substituting $S$ and $S^+$, is equivalent to $\frac{\|x\|_2^4}{\|c\|_2^2} \leqslant \frac{\|x\|_2^4}{\|\mathcal{R}(x)\|_1^2}$ or $\|c\|_2 \leqslant \|\mathcal{R}(x)\|_1$. The last inequality immediately holds since $\|c\|_2 \leqslant \|c\|_1$ and $\|c\|_1 \leqslant \|\mathcal{R}(x)\|_1$ by the triangle inequality. $\quad\square$

Similarly to DRIVE, one can deterministically set $S^+ = \frac{\|x\|_2^2}{\mathbb{E}\left[\|c\|_2^2\right]}$ and gain a $\frac{\pi}{2} - 1$ upper bound on the vNMSE for any dimension $d$. Notice that this requires calculating this expression; for any dimension $d$, one can approximate this quantity once to within arbitrary precision using a Monte-Carlo simulation. Nevertheless, as mentioned, this is less favorable as it introduces undesirable computational overhead for non uniform rotations.

### C.4 Implementation Notes

We note that it is possible to introduce a tradeoff between the additional complexity introduced by the computation of the centroids and the reduction in vMNSE compared to DRIVE. This is achieved by initializing the centroids symmetrically to that of DRIVE and continuing to apply iterations of Lloyd's algorithm. Each such additional iteration introduces more calculations but reduces the SSE.

In high dimensions, we find that the SSE of DRIVE and DRIVE$^+$ similar for vectors whose distribution admits finite moments (they converge to the same value). In low dimensions, we often find that after 2-3 such iterations, further improvement is negligible. Since Lloyd's algorithm admits an efficient GPU implementation, these extra iterations may offer a better vMNSE to computation tradeoff than optimally finding the centroids using less GPU-friendly implementations.

# D Supplemental Results for Structured Random Rotations

We next provide supplementary results for DRIVE and DRIVE$^+$ with a structured random rotation.

## D.1 Convergence of the Rotated Coordinates to a Normal Distribution

For completeness, we restate Lemma 4:

**Lemma 4.** *For all $i$, $\mathcal{R}_H(x)_i$ converges to a normal variable:* $\sup_{x \in \mathbb{R}} |F_{i,d}(x) - \Phi(x)| \leqslant \frac{0.409 \cdot \rho}{\sigma^3 \sqrt{d}}$.

*Proof.* By definition $\frac{1}{\sqrt{d}} \cdot \mathcal{R}_H(x)_i = \frac{1}{d} \cdot \sum_{j=1}^{d} x_j H_{ij} D_{jj}$ for all $i$. Due to the sign symmetry of $D_{jj}$, it holds that $x_j H_{ij} D_{jj}$ are zero-mean i.i.d. random variables. Also, $\mathbb{E}\left[x_j H_{ij} D_{jj}\right]^2 = \sigma^2$ and $\mathbb{E}\left[|x_j H_{ij} D_{jj}|\right]^3 = \rho$. Now, denote by $F_{i,d}$ be the cumulative distribution function (CDF) of $\frac{1}{\sigma} \cdot \mathcal{R}_H(x)_i$. Using the Berry–Esseen theorem [71], we obtain than $\sup_{x \in \mathbb{R}} |F_{i,d}(x) - \Phi(x)| \leqslant \frac{0.409 \cdot \rho}{\sigma^3 \sqrt{d}}$, where $\Phi$ the CDF of the standard normal distribution. This concludes the proof. $\qquad\square$

## D.2 Analysis of the 4th Moment When Using a Structured Random Rotation

Further assume that $\mathbb{E}[x_j^4] = M_4 < \infty$. Then, we have that,

$$\mathbb{E}\left[(\mathcal{R}_H \cdot x)_{i_1} \cdot (\mathcal{R}_H \cdot x)_{i_2} \cdot (\mathcal{R}_H \cdot x)_{i_3} \cdot (\mathcal{R}_H \cdot x)_{i_4}\right]$$

$$= \frac{1}{d^2} \sum_{j=1}^{d} \sum_{k=1}^{d} \sum_{l=1}^{d} \sum_{m=1}^{d} \mathbb{E}\left[x_j H_{i_1 j} D_{jj} \cdot x_k H_{i_2 k} D_{kk} \cdot x_l H_{i_3 l} D_{ll} \cdot x_m H_{i_4 m} D_{mm}\right]$$

$$= \frac{1}{d^2} \sum_{j=1}^{d} \sum_{k=1}^{d} \sum_{l=1}^{d} \sum_{m=1}^{d} H_{i_1 j} H_{i_2 k} H_{i_3 l} H_{i_4 m} \cdot \mathbb{E}\left[x_j x_k x_l x_m \cdot D_{jj} D_{kk} D_{ll} D_{mm}\right].$$

There are only two cases for which the expectation is not zero. Where there are two different indexes (where the resulting expectation is $\sigma^4$) and where all four are identical (where the resulting expectation is $M_4$). Therefore,

$$\mathbb{E}\left[(\mathcal{R}_H \cdot x)_{i_1} \cdot (\mathcal{R}_H \cdot x)_{i_2} \cdot (\mathcal{R}_H \cdot x)_{i_3} \cdot (\mathcal{R}_H \cdot x)_{i_4}\right]$$

$$= \frac{1}{d^2} \cdot \sigma^4 \cdot \left[\sum_{j=1}^{d} H_{1+(i_1-1)\oplus(i_2-1),j} \cdot \left[\sum_{k=1,k\neq j}^{d} H_{1+(i_3-1)\oplus(i_4-1),k}\right]\right]$$

$$+ \frac{1}{d^2} \cdot \sigma^4 \cdot \left[\sum_{j=1}^{d} H_{1+(i_1-1)\oplus(i_3-1),j} \cdot \left[\sum_{k=1,k\neq j}^{d} H_{1+(i_2-1)\oplus(i_4-1),k}\right]\right]$$

$$+ \frac{1}{d^2} \cdot \sigma^4 \cdot \left[\sum_{j=1}^{d} H_{1+(i_1-1)\oplus(i_4-1),j} \cdot \left[\sum_{k=1,k\neq j}^{d} H_{1+(i_2-1)\oplus(i_3-1),k}\right]\right]$$

$$+ \frac{1}{d^2} \cdot M_4 \cdot \sum_{j=1}^{d} H_{1+(i_1-1)\oplus(i_2-1)\oplus(i_3-1)\oplus(i_4-1),j}$$

Now we can calculate all the fourth moments.

There are four cases to consider: (1) for $i_1 = i_2 = i_3 = i_4$ we obtain $3 \cdot \sigma^4 \cdot \frac{d \cdot (d-1)}{d^2} + \frac{d}{d^2} \cdot M_4 = 3 \cdot \sigma^4 + O(\frac{1}{d})$; (2) for all pair equalities (e.g., $i_1 = i_2$ and $i_3 = i_4$ but $i_1 \neq i_2$) we obtain $\sigma^4 \cdot \frac{d \cdot (d-1)}{d^2} + \frac{d}{d^2} \cdot M_4 = \sigma^4 + O(\frac{1}{d})$; (3) when having a unique entry and at least a pair that equals (e.g., $i_1 = i_2, i_2 \neq i_4$ and $i_3 \neq i_4$ regardless of whether $i_2 = i_3$) we immediately obtain zero (recall that all odd moments are zero); (4) for all distinct entries it may be the case that $1 + (i_1-1) \oplus (i_2-1) \oplus (i_3-1) \oplus (i_4-1) = 1$, therefore we obtain $\frac{d}{d^2} \cdot M_4 = O(\frac{1}{d})$.

These indeed approach, with a rate of $\frac{1}{d}$, the 4'th moments of *independent* normal random variables.

# E   Additional Simulation Details and Results

In this section we provide additional details and simulation results. The source code and full reproduction instructions are in the supplementary material archive.

## E.1   Preexisting Code and Datasets Description

**Code.** Our federated learning experiments use the TensorFlow Federated [45] library with the Hadamard rotation and the Kashin's representation code taken from the TensorFlow Model Optimization Toolkit [72]. Tensorflow and its subpackages are licensed under the Apache License, Version 2.0 [73]. The rest of the experiments use PyTorch [44], which is licensed under a BSD-style license [74].

**Datasets.** Next we provide a quick overview of the datasets used:

*MNIST* [50, 51]. Includes 70,000 examples of $28 \times 28$ grayscale labeled images of handwritten digits (10 classes).

*EMNIST* [52]. Extends MNIST and includes 749,068 examples of $28 \times 28$ grayscale labeled images of 62 handwritten characters.

*CIFAR-10* and *CIFAR-100* [53]. Both consist of 60,000 examples of $32 \times 32$ images with 3 color channels, in each dataset the images are labeled into 10 and 100 classes, respectively.

*Shakespeare* [54]. Contains 18,424 examples, where each example is a contiguous set of lines spoken by a character in a play by Shakespeare.

*Stack Overflow* [55]. Consists of 152,404,765 examples, each containing the text of either an answer or a question. The data consists of the body text of all questions and answers

For the federated learning experiments, following previous works in the field [1, 60], the EMNIST, Shakespeare, and Stack Overflow datasets are partitioned naturally to create an heterogeneous unbalanced client dataset distribution. Specifically, they are partitioned by character writer, play speaker, and Stack Overflow user, respectively. In the CIFAR-100 federated experiment, the examples are split among 500 clients using a random heterogeneity-inducing process described in [61, Appendix C.1].

All datasets were released under CC BY-SA 3.0 [75] or similar licenses, the exact details are in the cited references.

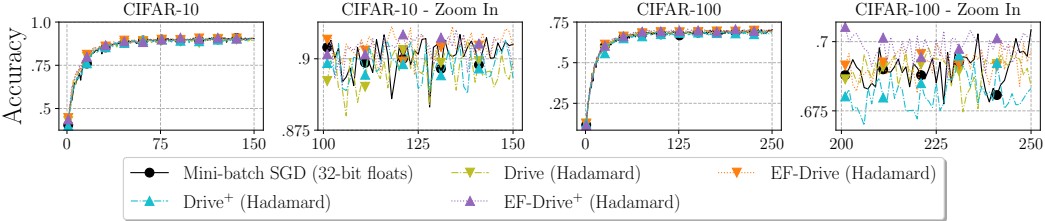

Figure 4: Distributed CNN EF experiments comparing DRIVE and DRIVE$^+$ with and without EF. Accuracy per round on distributed learning tasks, with a zoom-in on the last 50 rounds.

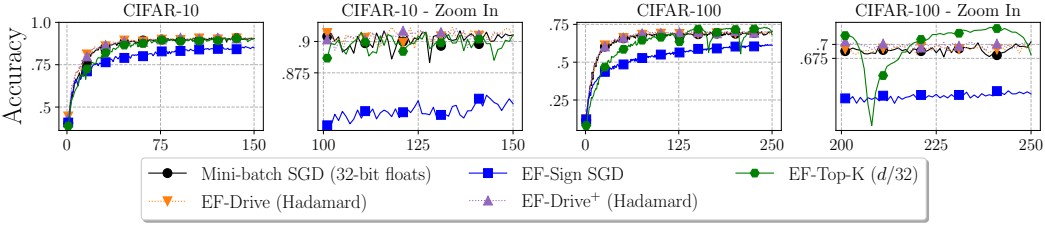

Figure 5: Distributed CNN EF experiments comparing DRIVE and DRIVE$^+$ with and techniques utilizing EF. Accuracy per round on distributed learning tasks, with a zoom-in on the last 50 rounds.

## E.2 Error-Feedback Experiments

We next conduct error feedback (EF) experiments for the distributed CNN training. For DRIVE (and similarly for DRIVE$^+$), we use the following formula to set the scale, $S = \min\{2 \cdot \frac{\|\mathcal{R}_H(x)\|_1}{d}, \frac{\|x\|_2^2}{\|\mathcal{R}_H(x)\|_1}\}$. This scale ensures that the *compressor assumption* [65, Assumption A] holds but tries to make use of the "unbiased" scale whenever possible. In fact, in our CNN simulations, we did not encounter a single iteration in which $2 \cdot \frac{\|\mathcal{R}_H(x)\|_1}{d} \leqslant \frac{\|x\|_2^2}{\|\mathcal{R}_H(x)\|_1}$.

Figure 4 shows that with and without EF, DRIVE and DRIVE$^+$ result in similar performance. Also, as evident in Figure 5, DRIVE and DRIVE$^+$ show favorable performance in comparison to EF-Sign SGD and Top-K. Here, we configured the Top-K algorithm with $K = d/32$; note that this requires more than one bit per coordinate, as the sender needs to encode the indices of the sent coordinates in addition to their values.

## E.3 Additional vNMSE-Speed Tradeoff Details and Results

**Experiment Configuration.** The experiments were executed over Intel Core i9-10980XE CPU (18 cores, 3.00 GHz, and 24.75 MB cache), 128 GB RAM, NVIDIA GeForce RTX 3090 GPU, Ubuntu 20.04.2 LTS operating system, and CUDA release 11.1, V11.1.105. We draw 100 vectors[3] from a Lognormal(0,1) distribution and measure 100 encodings of each vector. Similarly to Figure 1, each of $n = 10$ clients are given the same vector at each iteration. Due to its high runtime, we only evaluate the uniform random rotation on the low ($d \in \{128, 8192\}$) dimensions.

**Experiment Results Summary.** The results, given in Table 1, show that DRIVE and DRIVE$^+$ are the most accurate algorithms and their NMSE converges to the theoretical NMSE of $\frac{\pi/2-1}{10} \approx 0.0571$ even in dimensions as small as $2^{13}$. In even lower dimensions, such as $d = 128$, DRIVE$^+$ is more accurate than DRIVE, albeit slightly slower. Running DRIVE and DRIVE$^+$ with uniform rotation instead of Hadamard yield even lower NMSE, but also takes more time. DRIVE is also almost as fast as Hadamard with 1-bit SQ [2] and is 12x faster than Kashin with 1-bit SQ. We conclude that DRIVE with Hadamard offers the best tradeoff in all cases, except when the dimensions is small, in which case one may choose to use DRIVE$^+$ and possibly uniform random rotation.

| Dimension ($d$) | Hadamard + 1-bit SQ | Kashin + 1-bit SQ | Drive (Uniform) | Drive$^+$ (Uniform) | Drive (Hadamard) | Drive$^+$ (Hadamard) |
|---|---|---|---|---|---|---|
| 128 | 0.5308, *0.93* | 0.2550, *6.06* | 0.0567, *14.95* | **0.0547**, *18.25* | 0.0591, *0.98* | 0.0591, *2.07* |
| 8,192 | 1.3338, *1.53* | 0.3180, *10.9* | **0.0571**, *2646* | **0.0571**, *2672* | **0.0571**, *1.58* | **0.0571**, *2.93* |
| 524,288 | 2.1456, *3.76* | 0.3178, *30.8* | — | — | **0.0571**, *3.78* | **0.0571**, *5.72* |
| 33,554,432 | 2.9332, *40.9* | 0.3179, *1714* | — | — | **0.0571**, *43.0* | **0.0571**, *252* |

Table 3: A *commodity machine* comparison of empirical NMSE and average per-vector encoding time (in milliseconds) for distributed mean estimation with $n = 10$ clients (same as in Figure 1) and Lognormal(0,1) distribution. Each entry is a (NMSE, *time*) tuple and the best result is highlighted in **bold**.

**Speed Measurements on a Commodity Machine.** The experiment in Table 1 in the paper was performed on a high-end server with a NVIDIA GeForce RTX 3090 GPU. For completeness, we also run measurements on a commodity machine. Specifically, we re-execute the experiments over Intel Core i7-7700 CPU (8 cores, 3.60 GHz, and 8 MB cache), 64 GB RAM, NVIDIA GeForce GTX 1060 (6GB) GPU, Windows 10 (build 18363.1556) operating system and CUDA release 10.2, V10.2.89. The results, shown in Table 3 show a similar trend. Again, as expected, DRIVE and DRIVE$^+$ with uniform rotation are more accurate for small dimensions but are significantly slower. On this machine, DRIVE is $6.18\times$-$39.8\times$ faster than Kashin and, again, about as fast as Hadamard. All NMSE measurements, as expected yielded the same results as in Table 1.

---

[3]except for $d = 2^{25}$, where we use 10 encodings of each vector due to the fact that in such high dimension, random vectors are concentrated close to their expectation.

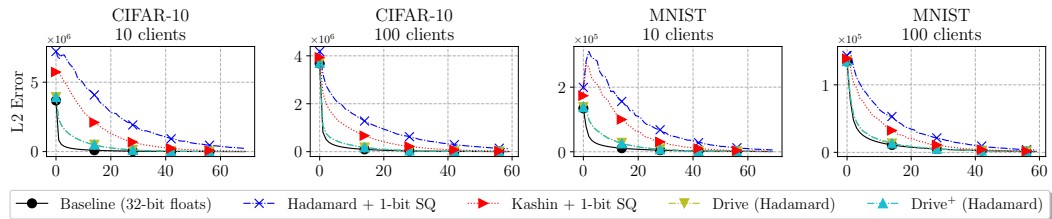

Figure 6: L2 error per round for distributed power iteration.

## E.4 Federated Learning Experiments Configuration

The experiments were executed over a university cluster that contains several types of NVIDIA GPUs (TITAN X (Pascal), TITAN Xp, GeForce GTX 1080 Ti, GeForce RTX 2080 Ti, and GeForce RTX 3090) with no specific hardware guarantee.

| Task | Clients per round | Rounds | Batch size | Client lr | Server lr |
|------|-------------------|--------|------------|-----------|-----------|
| EMNIST | 10 | 1500 | 20 | 0.1 | 1 |
| CIFAR-100 | 10 | 6000 | 20 | 0.1 | 0.1 |
| Shakespeare | 10 | 1200 | 4 | 1 | 1 |
| Stack Overflow | 50 | 1500 | 16 | 0.3 | 1 |

Table 4: Summary of main hyperparameters for federated learning tasks.

Table 4 contains a summary of the hyperparameters we used. Those generally correspond to the best setup described in [61, Appendix D] for FedAvg except for the following changes: (1) We increased the number of rounds of the EMNIST task from 4000 to 6000 to show a clear convergence; (2) We used a server momentum of 0.9 (named *FedAvgM* in [61]) for the Stack Overflow task since otherwise, this task produces noisy results due to its extreme unbalancedness in the number of samples per client; and (3) For FetchSGD on EMNIST and Stack Overflow we set the server learning rate to 0.3 as it offered improved performance.

We compress each layer separably due to the implementation of the libraries we use and algorithms we compare to (specifically, see "CLASS ENCODEDSUMFACTORY" [76] in TensorFlow Federated). This layer-wise compression reduces the dimension in practice and explains the difference seen between DRIVE and DRIVE$^+$ performance. Additionally, all compression algorithms skip layers with less than 10,000 parameters since those are usually either normalization or last fully-connected layers, which are more cost-effective to send as-is.

## E.5 Distributed Learning Experiments Configuration

The experiments were executed over Intel Core i9-10980XE CPU (18 cores, 3.00 GHz, and 24.75 MB cache), 128 GB RAM, NVIDIA GeForce RTX 3090 GPU, and Ubuntu 20.04.2 LTS OS.

For the distributed CNN training we have used an SGD optimizer with a Cross entropy loss criterion, a momentum of 0.9, and a weight decay of 5e-4. The learning rate of all algorithms was set to 0.1 for the CIFAR-10 over ResNet-9 experiment and 0.05 for the CIFAR-100 over ResNet-18 experiments. TernGrad was an exception and its learning rate was set to 0.01 and 0.005 as it offered improved performance. The batch size in all experiments and clients was set to 128.

## E.6 K-Means and Power Iteration Simulation Results

**Distributed Power Iteration.** Power Iteration is a simple method for approximating the dominant Eigenvalues and Eigenvectors of a matrix. It is often used as a sub-routine in more complicated tasks such as Principal Component Analysis. In our distributed setting, at each epoch, the server broadcasts the current estimated top eigenvector to all clients. Then, each client: (1) updates the top eigenvector based on its local data; (2) computes its diff to the current estimated top eigenvector; (3) compresses the diff vector; (4) sends it to the server, possibly scaled by a *learning rate* to ensure

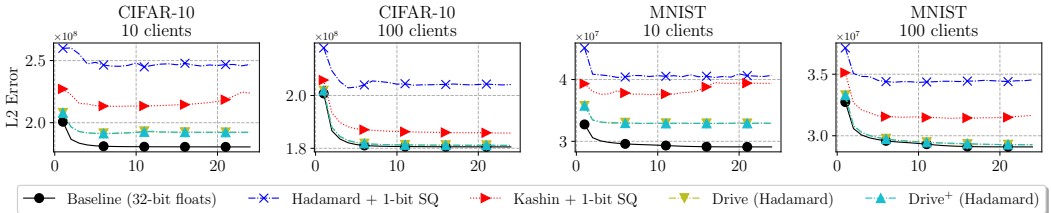

Figure 7: L2 error per round for distributed K-Means.

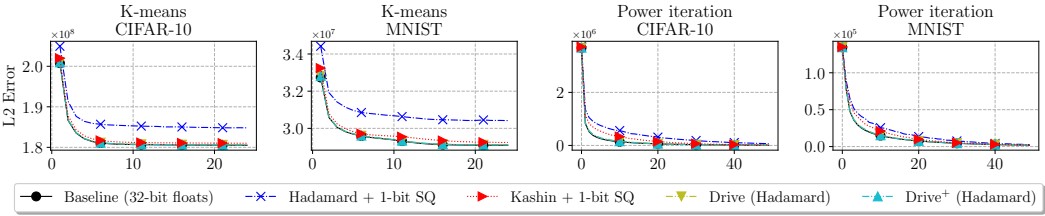

Figure 8: Distributed K-Means and power iteration of CIFAR-10 and MNIST with 1000 clients.

convergence to due the high estimation variance. Figure 6 presents the results in a setting with $n = 10$ and $n = 100$ clients and a learning rate of 0.1.

In both datasets, and for both $n = 10$ and $n = 100$ clients, DRIVE and DRIVE$^+$ have lower error and faster convergence than other compression algorithms. Our convergence is faster when using a larger number of clients, which matches the theoretical analysis which shows that the estimates of DRIVE (Hadamard) and DRIVE$^+$ (Hadamard) are (nearly) unbiased for such a high dimension.

**Distributed K-Means.** In this distributed learning task, the server coordinates a run of Lloyd's algorithm over a distributed dataset. At each training round, the server broadcasts the centroids' coordinates to the clients. Then, each client: (1) updates these coordinates based on its local data; (2) compresses them; (3) sends them to the server. The centroid weights (i.e., the number of observations assigned to each centroid) require fewer bits and are essential for K-Means to converge quickly; therefore, the weights are sent without compression.

Figure 7 presents the results with $n = 10$ and $n = 100$ clients. As shown, DRIVE and DRIVE$^+$ have a lower error than other compression algorithms. While none of the compression methods is competitive with the baseline when using $n = 10$ clients, DRIVE and DRIVE$^+$ are the only algorithms that have comparable accuracy for $n = 100$.

**Increasing the Number of Clients.** Figure 8 depicts simulation results for the distributed K-Means and distributed Power Iteration experiments with $n = 1000$ clients. The results indicate a similar trend. DRIVE and DRIVE$^+$ offer the lowest L2 error among the low-bandwidth techniques, and they are now closer to the uncompressed baseline due to the larger number of clients, which further reduces the estimation variance.

### E.7 Comparison With Entropy Encoding

We conduct an experiment for distributed mean estimation with $n = 10$ clients (same as in Figure 1 and Table 1) with the Lognormal(0,1) distribution. We determine how many bits per coordinate are required by a compression method that uses stochastic quantization followed by entropy encoding to reach the same NMSE as that of DRIVE (that uses a single bit per coordinate). We examine two methods: (1) standard stochastic quantization followed by Huffman encoding; (2) stochastic quantization with levels chosen for entropy encoding followed by Huffman encoding, as proposed in [2] (see Section 4). We refer to this second variant as Enhanced SQ. We introduced an increasing number of quantization levels for each of these methods until the resulting empirical NMSE was similar to that of DRIVE. We repeat each experiment 100 times and report the mean value. The results summarized in Table 5 indicate that these methods require at least 1.261 bits per coordinate.

| Dimension ($d$) | DRIVE (Hadamard) NMSE (1-bit) | Required bits/coordinates | |
| --- | --- | --- | --- |
| | | SQ + Huffman | Enhanced SQ + Huffman [2] |
| 128 | 0.0591 | 1.284 | 1.261 |
| 8,192 | 0.0571 | 1.335 | 1.295 |
| 524,288 | 0.0571 | 1.324 | 1.298 |
| 33,554,432 | 0.0571 | 1.321 | 1.301 |

Table 5: Empirical NMSE for distributed mean estimation with $n = 10$ clients (same as in Figure 1 and Table 1) with the Lognormal(0,1) distribution. We find the number of bits per coordinate that are required by the entropy encoding methods to reach the same empirical NMSE as that of DRIVE (that uses a single bit per coordinate).

There are other uses of entropy encoding techniques for machine learning tasks as suggested by [2, 9]. Such methods often introduce higher computational costs. We leave further comparisons of DRIVE and entropy encoding for specific problems for future work.

# F   Lower Bounds

It is proven in [14] that *any* unbiased algorithm must have a 1b - Vector Estimation vNMSE of at least $\frac{1}{3}$ and any biased algorithm incurs a vNMSE of at least $\frac{1}{4}$. However, it does not appear that their proof accounts for algorithms in which Buffy and Angel may have shared randomness, as is the case for all algorithms proposed in our paper. The reason is that in [14] it is assumed that if Buffy sends $b$ bits, Angel only has $2^b$ possible estimates, one for each of the $2^b$ possible messages. However, if shared randomness is allowed, there are more possible values for $\hat{x}$. In fact, if the number of shared random bits is not restricted, the number of different possible estimates can be unbounded. Additionally, it is shown in [7] that for a single number ($d = 1$), using shared randomness allows algorithms to have an MSE lower than the optimal algorithm when no such randomness is allowed.

Nonetheless, their result implies that any algorithm that uses $O(d)$ shared random bits (as does our Hadamard-based variants) has a vNMSE of $\Omega(1)$, i.e., DRIVE and DRIVE$^+$ are asymptotically optimal. The reason is that Buffy could generate (private) random bits and send them as part of the message to mimic shared random bits. For example, in our Hadamard variant, Buffy could send the random diagonal matrix $D$ using $d$ additional bits. This implies that any biased algorithm with at most $d$ shared random bits must have a vNMSE of at least $\frac{1}{16}$ and any unbiased solution must have a vNMSE of at least $\frac{1}{15}$.