# OpenReview forum: "DRIVE: One-bit Distributed Mean Estimation"
_NeurIPS.cc/2021/Conference — NeurIPS 2021 Poster_

### Official Review · Reviewer_sJg9 · 2021-07-16

**Rating:** 7
**Confidence:** 4

**Summary:**

This paper studies the problem of distributed mean estimation under a single bit (per coordinate) communication constraint. It proposes a novel compression scheme based on random rotation and sign quantization. The suggested method is applied to distributed and federated machine learning problems and shows improved results over previously studied quantization methods.

**Limitations And Societal Impact:**

Limitations of the work were mostly addressed. Negative societal impact concerns are not applicable due to the theoretical nature of the paper.

**Main Review:**

This work proposes and analyzes a compression method that is based on multiplication by rotation matrix followed by deterministic sign quantization and scaling. To make it scalable and computationally efficient for large dimensions, the authors suggest using structured randomized Hadamard transform, which is not a novel thing on its own.
The paper is clearly written, but it can benefit from adding a table with a summary of the obtained theoretical results and comparison to previous approaches.

Theorem 5 is a simple consequence of the known fact
$$
\mathrm{E} \left\Vert \frac{1}{n} \sum_{i=1}^n \left(X_i - \mathrm{E}\left[X_i\right]\right) \right\Vert_2^2 = \frac{1}{n^2} \sum_1^n \mathrm{E} \lVert X_i - \mathrm{E}\left[X_i\right] \rVert_2^2,
$$
for any independent random variables $X_i$. This was previously used in many analyzes of distributed optimization methods with compressed (gradient) updates.

 As soon as DRIVE with a structured random rotation cannot be made unbiased, I would like to point out the authors to the fact that distributed/federated optimization methods with biased compressors are not guaranteed to converge in the general case of convex functions and may even diverge exponentially [1]. That is why the Error-Feedback mechanism [2, 3] is required to guarantee the theoretical foundation of the proposed method in the optimization setting. In my opinion, this is the main weak theoretical point of the paper, but it can be fixed without a great effort with additional experiments.

Regarding the experiments. Could you please comment why your experimental results contradict with findings of Caldas et al., which showed a better performance of Kashin representation in comparison to Hadamard transform? I think that more discussion regarding this would be helpful and insightful.


**Minor issues/comments:**

I would like to ask the authors to define (in the paper) the term PRNG, which was used on pages 2, 4 in the context of shared randomness.

It seems to me that the paper can benefit from moving some (technical) proofs into Appendix. For instance, the proofs of Theorem 1 and Lemma 1 are pretty simple, and their idea can be expressed in one sentence. This can free the space for a more detailed description of DRIVE$^+$.


[1] Beznosikov A., et al. “On biased compression for distributed learning”. arXiv preprint arXiv:2002.12410, 2020.

[2] Sebastian U. Stich, et al. “Sparsified SGD with memory”. In Advances in Neural Information Processing Systems (NeurIPS), 2018.

[3] Richtárik P., et al. “EF21: A New, Simpler, Theoretically Better, and Practically Faster Error
Feedback”. arXiv preprint arXiv:2106.05203, 2021.

[4] Caldas S., et al. “Expanding the Reach of Federated Learning by Reducing Client Resource Requirements”. arXiv preprint arXiv:1812.07210, 2018.

**Time Spent Reviewing:**

7

---

> ### Author Response · Authors · 2021-08-09
> **Author response**
>
> Thank you for your comments.
>
> We will add a summary table as you suggested.
> We also agree that the proof of Theorem 5 is straightforward and debated whether to put it in the main text. Eventually, we figured that it’s better to have it there (as we expect some number of readers would not be aware of it), as our main goal was to get bounds on the distributed mean estimation problem.
> We will clarify in the text that, without additional assumptions (as explained in Section 6) on the gradient distribution, error feedback is necessary to ensure convergence and recover the converge rate of non-compressed SGD.
> We will also add an EF experiment.
>
>
> We do not believe that there is a contradiction between our experimental results and the findings of Caldas et al.
> In line with the findings of Caldas et al., using the same quantization method (stochastic quantization), we observed that Kashin + 1-Bit SQ is more accurate (albeit a bit slower) than Hadamard + 1-Bit SQ.
> In DRIVE, our observation was that the Hadamard transform converges, for high dimensions, to a fixed distribution. Specifically, all rotated coordinates follow the same (converging to normal) distribution, thus allowing us to optimize the quantization.
> In contrast, Kashin’s representation can result in differently distributed coordinates. Therefore, we showed that by using DRIVE’s quantization on top of Hadamard, one could get a lower error than using Kashin with *stochastic quantization*.
>
> In summary, the reason that there is no contradiction is we are using a different, tuned quantization (DRIVE’s quantization vs. stochastic quantization), and this quantization applies to the distribution induced by the Hadamard transform but not to the Kashin representation. We will clarify this in the text.

---

> > ### Comment · Reviewer_sJg9 · 2021-08-23
> > **Some requests and recommendations**
> >
> > Thank you for your clarifications.
> >
> > I support the point of Reviewer d5Wo of moving some of the experimental results to the Supplementary Material section. For now, this part indeed seems quite heavy.
> >
> > Regarding the necessity of some kind of Error-Feedback mechanism for convergence guarantees. Can you show that DRIVE with a Structured Random Rotation actually satisfies the relative "bounded variance" assumption
> > $$
> > \mathbb{E} \||x - \hat{x}\||_2^2 \leq \delta \||x\||_2^2  \qquad \forall x \in \mathbb{R}^d,
> > $$
> > that is required to use Error-Feedback?
> >
> > I would like the authors to include Accuracy/Time plots for distributed learning tasks as quantization methods do not support Al-Reduce, which is a more communication-efficient aggregation protocol. Due to this, compression can even lead to a slow down of the training process [1, 2]. That is why I think it is important to show the benefits of the proposed approach on a more realistic scale and mention it among limitations.
> >
> > ***
> >
> > [1] Xu H., et al. “Compressed communication for distributed deep learning: Survey and quantitative evaluation”. [Technical report](https://repository.kaust.edu.sa/handle/10754/662495), 2020.
> >
> > [2] Agarwal S., et al. “On the Utility of Gradient Compression in Distributed Training Systems”. arXiv preprint arXiv:2103.00543, 2021.

---

> > > ### Author Response · Authors · 2021-08-24
> > > **Author response**
> > >
> > > Thank you for pointing out these observations and suggestions.
> > >
> > > As suggested, we will move some of the experimental results to the supplementary material to make room for addressing the reviews.
> > >
> > > DRIVE with a Structured Random Rotation satisfies the relative "bounded variance" assumption according to Lemma 3 (with $\delta=0.5$). A slight extension of Lemma 3 indicates that any scale that satisfies $\frac{||\mathcal R_H(x) ||_1}{d} \le S < 2 \cdot \frac{|| \mathcal R_H(x) ||_1}{d}$ suffices, which allows further optimization for EF by choosing the best scale. We will clarify this in the text.
> > >
> > > We are now in the process of conducting an EF experiment as requested by the reviewers. Initial results indicate the same gains we already displayed in the paper (i.e., without EF).
> > >
> > > We agree that quantization techniques may introduce overhead in the context of All-Reduce (depending on the network architecture and communication patterns). However, as mentioned by the reviewer, this observation concerns the general approach of compressing gradient vectors and is not specific to DRIVE. We will clarify this observation in the limitations section.
> > >
> > > The future research direction of optimizing communication libraries and network architectures to maximize the benefits quantization may offer is timely and challenging. Most recent studies already address these challenges and propose frameworks that efficiently support quantized gradient communications (e.g., [1]), offering lower wall-clock training time.  That is, even with the existence of All-Reduce, we expect (and [1] provides backing for the idea) that gradient compression will continue to be valuable in this context, and certainly others;  our goal with DRIVE is precisely to offer better compression.
> > >
> > > Specifically for DRIVE, we will add a brief discussion in the limitations section with possible challenges in the context of All-Reduce and future directions for minimizing potential overheads (e.g., bucketize co-located workers and apply DRIVE's quantization only for cross-rack traffic).
> > >
> > >
> > > [1] Youhui Bai, Cheng Li, Quan Zhou, Jun Yi, Ping Gong, Feng Yan, Ruichuan Chen, Yinlong Xu, Gradient Compression Supercharged High-Performance Data Parallel DNN Training, To appear in the proceedings of the 28th ACM Symposium on Operating Systems Principles (SOSP), 2021.

---

### Official Review · Reviewer_UVT6 · 2021-07-16

**Rating:** 6
**Confidence:** 1

**Summary:**

The paper considers the problem of communication-efficient distributed mean estimation. This paper proposes a novel algorithm DRIVE which uses as little as almost 1 bit for each coordinate of the vector and estimate the distributed mean with bounded error. The idea is to use a uniform random rotation to minimize the vNMSE. Besides, DRIVE can use structured random rotation to accelerate the compression for high dimensional vectors. The empirical evaluation of DRIVE shows that federated learning with gradients compressed by DRIVE can achieve faster convergence than other compression schemes.

**Limitations And Societal Impact:**

Yes.

**Main Review:**

Overall, the writing of the paper is clean and easy to understand. As a majority of the evaluation is done for federated learning task, it would be better to provide a convergence result for DRIVE+SGD to show the impact of error of distributed mean estimation on the rates. Besides, there is a recent work (Davies et al., 2020) could be an interesting reference.

Davies P, Gurunathan V, Moshrefi N, et al. New Bounds For Distributed Mean Estimation and Variance Reduction[J]. arXiv preprint arXiv:2002.09268, 2020.

**Time Spent Reviewing:**

5

---

> ### Author Response · Authors · 2021-08-09
> **Author response**
>
> Thank you for your comments.
>
> If one uses DRIVE with a uniform random rotation, the encoding is unbiased and the convergence is guaranteed at the same asymptotic rate as without compression. When using DRIVE with the Hadamard transform, additional assumptions are required (see Section 6).
> Otherwise, as suggested by reviewer sJg9, error feedback (EF) can be used to guarantee the recovery of the convergence rate. We will clarify this in the text and add an EF experiment.
>
> Thank you for pointing out the work of Davies et al.
> We will relate our work to it in the text.

---

### Official Review · Reviewer_QoQf · 2021-07-17

**Rating:** 5
**Confidence:** 4

**Summary:**

This paper studies the distributed mean estimation. In this problem, there are n clients,  each of which has a d-dimensional real-valued; the goal is to estimate their mean on the sever side using minimum communication cost. The paper focuses on 1-bit quantization methods, i.e., each client sends one bit per coordinate and d(1+o(1)) bits in total. The algorithms are standard (including the Hadamard transformation based method); the main difference is new scaling factors S. Theoretical analysis on the expectation and variance are provided. Empirical results also show some advantages of the new estimate.

**Limitations And Societal Impact:**

Yes

**Main Review:**

Strength:
1. The problem studied is fundamental and well-motivated.
2. Theoretical guarantees are proved.
3. Empirical results show that the new estimate achieve improved MSE among 1-bit quantization methods.

Weakness:
1. All algorithms mostly use existing techniques. The
2. The analyses are not too difficult.
3. My main concern is that the paper only focus on 1-bit quantization, which I think is somewhat restricted. 1-bit quantization is often suboptimal; multi-level quantization could be much better, see e.g., qsgd [1]. In particular, when the vectors are sparse or skew, the budget for different  coordinates should be different. In particular [2] shows that the total communication cost can be sublinear in d for skew vectors. Moreover, the random rotation operation applied on the original vectors destroys such sparsity/skewness. Such skewness is bad for 1-bit quantization, which is why random rotation is applied to make the vector “flat”. But skewness can actually save communication provably, if multi-level quantization is used.
4. In the experiments, multi-level quantization QSGD, variable-length coding methods [3], etc. should also be compared.




[1] D. Alistarh, D. Grubic, J. Li, R. Tomioka, and M. Vojnovic. Qsgd: Communication-efficient sgd via gradient quantization and encoding.\
[2] Z. Huang, Y. WANG, K. Yi, et al. Optimal sparsity-sensitive bounds for distributed mean estimation.\
[3] A. T. Suresh, X. Y. Felix, S. Kumar, and H. B. McMahan. Distributed mean estimation with limited communication.


**Time Spent Reviewing:**

4

---

> ### Author Response · Authors · 2021-08-09
> **Author response**
>
> Thank you for your comments.
>
> Comment 3:
> As we note in the paper: “We note that TernGrad is a low-bit variant of a well-known algorithm called \emph{QSGD}~\cite{NIPS2017_6c340f25}, and we use TernGrad since we found it to perform better in our experiments”.
> That is, in all settings, we have seen TernGrad does slightly better than QSGD, and therefore QSGD was omitted from the evaluation.
> In fact, when restricted to two quantization levels, TernGrad is identical to QSGD’s max normalization variant with clipping; we will clarify it in the text.
>
> You are also correct that when the data is sparse or skewed (or otherwise structured), one may use other techniques. We will add a clarification about that in the text.
>
> Comment 4:
> By using multiple quantization levels, the encoding may not result in just $d(1+o(1))$ bits for general vectors, even with variable-length encoding (as the entropy is more than one bit due to the sign-symmetry). We will clarify that when multiple bits per coordinate are allowed, one may use variable-length encoding techniques as suggested in QSGD.
> We ran additional experiments with more bits using $d=2^{25}$ and ten clients (the same setting as the large dimension in table 1).
> The result shows that Hadamard + 2-Bit SQ (i.e., taking $2d(1+o(1))$ bits uncompressed) has an NMSE of 0.257, about five times larger than DRIVE with a single bit. We note that variable length encoding does not reduce the encoding length back to $d(1+o(1))$ bits.
> QSGD’s performance (with L_inf normalization) is further behind and has an NMSE of 5.05.
> Kashin is more competitive with a larger number of bits.
> Kashin + 2-Bit SQ (i.e., with $2.34d(1+o(1))$ bits) has an NMSE of 0.02727. While this is lower than DRIVE with one bit per coordinate, we ran DRIVE+ with $2d(1+o(1))$ bits (i.e., we used K=4 centroids, as suggested in the paper’s Conclusion) and measured an NMSE of 0.01331, i.e., an improvement of 51%. We note that when using more than two centroids, we can also use compression (such as arithmetic encoding) to reduce the representation length further (roughly, by 9% for $K$=4, and increasing for larger $K$), as done in variable length encoding in [3].
> We also note that while using a higher number of centroids reduces the error, we currently don’t have stronger error bounds; i.e., DRIVE with more than one bit per coordinate requires further research (as we state also in the paper).

---

### Official Review · Reviewer_d5Wo · 2021-08-01

**Rating:** 6
**Confidence:** 4

**Summary:**

The paper studies distributed mean estimation in the $d$-dimensional space where each client can use only $d(1+o(1))$ bits for communication. A crucial assumption is that the server and the $n$ clients can have shared randomness while each client can independently communicate with the server. Under such an assumption, the authors proposed two schemes and demonstrated the efficiency of their new approaches.

**Limitations And Societal Impact:**

The authors clearly stated the limitations of their work in the second paragraph of the Conclusion section. I do have some additional suggestions about the computational efficiency aspect. In Checklist, the authors indicated that "we do not believe there is an inherit (inherent?) negative societal impact in this work."

**Main Review:**

Several comments are in order.


1) Most parts of the paper read well with concise notations. In particular, I like the way of illustration that always starts with the Alice-Bob example. The proposed algorithms are also conceptually simple and easy to understand.


2) The authors propose two schemes: One utilizing uniformly random rotation matrices, and the other uses Hadamard transforms. The first scheme (DRIVE) seems computationally inefficient but works for all cases, and the second (DRIVE$^+$) is more efficient and works for scenarios with structural assumptions.


3) I wonder if the assumption of shared randomness is typical in the literature and practice (e.g., in Federated learning applications). In particular, the server needs to know the randomness of $\textbf{all}$ clients, and uniformly generating rotation matrices seems to require lots of random bits. It will be great if the authors can clarify this point with quantitative statements such as how many random bits are sufficient.


4) The $\textbf{only}$ approach with a satisfactory theoretical guarantee is DRIVE accompanied by uniformly random rotation matrices (Theorem 5). Here, I do have concerns about the efficiency of the algorithm.

     - On page 1, the authors mention that approaches via variable-length encoding have high computational costs. I thought that this hinted that the proposed schemes should be much more efficient.

     However, as Table 1 demonstrates, DRIVE with a uniformly drawn rotation matrix is $1{,}000$ times slower than some existing approaches.


5) I like the experimental results as they demonstrate that the proposed schemes lead to concrete improvements on standard test datasets. But I felt slightly overwhelmed by the list of highly compressed experiment descriptions, especially given that most relevant details appear in the supplementary. I suggest the authors pick a few experiments to include in the main paper with more information. This change should probably enhance the readability of the submission.


6) Another comment is about the competing algorithms. It seems that Hadamard + 1-bit SQ and Kashin + 1bit SQ are not designed for the $d(1+o(1))$ setting. I am slightly concerned about the fairness of comparison (e.g., lines 321-322). I wonder if the authors can add comparisons with Hadamard/Kashin + $c$-bit SQ for some $c>1$ as a relaxation since achieving $d(1+o(1))$ may not be crucial in many applications.


7) I appreciate the authors for being honest and providing a list of limitations of their work at the end of the submission. This part does not seem to mention the computational aspects of the algorithms. I think it might be good to state the time complexities of the proposed algorithms and some existing ones clearly at someplace.


8) Minor comments:
    - Is there a reason for normalizing the MSE of mean estimation by the sum of squared norms of the individual vectors (e.g., line 51)? I think it might be natural and more consistent if the normalizing factor is the squared norm of the mean.
   - Should one increase the spacing between lines 59 and 60?
   - Line 73: Is the word "it" referring to Alice?
   - Page 3: Should Algorithm 1 have a caption?
   - Line 227: Is there an actual application where 1.17 bits per coordinate is much worse than 1+o(1) bits?

**Time Spent Reviewing:**

5

---

> ### Author Response · Authors · 2021-08-09
> **Author response**
>
> Thank you for your comments.
>
> Comment 3:
> The shared randomness assumption seems standard (e.g., it is also used by the most relevant previous papers: Hadamard + SQ [2] and Kashin + SQ [11]).
> The reason is that shared randomness is easy to implement (e.g., we can use the round number and client ID as a shared pseudo-random number generator seed and generate bits from a pseudo-random number generator). Further, it is not too costly (e.g., the Hadamard transform only requires $d$ random bits while uniform rotation also takes $O(d)$ bits assuming finite precision).
>
> Comment 6:
> We have compared DRIVE to other algorithms with at least $d(1+o(1))$ bits. For example, Kashin + 1-Bit SQ requires $\approx1.17d(1+o(1))$ bits.
> We ran additional experiments with more bits using $d=2^{25}$ and ten clients (the same setting as the large dimension in table 1).
> The result shows that Hadamard + 2-Bit SQ (i.e., taking $2d(1+o(1))$ bits) has an NMSE of 0.257, about five times larger than DRIVE with a single bit.
> QSGD’s performance (with L_inf normalization) is further behind and has an NMSE of 5.05.
> Kashin is more competitive with a larger number of bits.
> Kashin + 2-Bit SQ (i.e., with $\approx2.34d(1+o(1))$ bits) has an NMSE of 0.02727. While this is lower than DRIVE with one bit per coordinate, we ran DRIVE+ with $2d(1+o(1))$ bits (i.e., we used $K=4$ centroids, as suggested in the paper’s Conclusion) and measured an NMSE of 0.01331, i.e., an improvement of 51%. We note that while using a higher number of centroids reduces the error, we currently don’t have stronger error bounds; i.e., DRIVE with more than one bit per coordinate requires further research (as we state also in the paper).
>
> Comment 7:
> You are correct that the structured-rotation-based techniques, including DRIVE, are more computationally expensive than linear-time solutions like TernGrad. We will add this clarification to the limitations discussion.
> That said, we believe that DRIVE’s encoding time is still short compared with the time it takes to compute the gradient before compression. Thus, this may not be a significant limitation. For example, we measured that it takes $\approx$470ms for computing the gradient on a RESNET18 architecture (for CIFAR100, batch size = 128, using GTX 1060) while the encoding takes $\approx$2.8ms in DRIVE (Hadamard). That is, the overall computation time is only increased by 0.6% while the error reduces significantly. Taking the transmission and model update times into consideration would reduce the importance of the compression time further.

---

### Decision · Program_Chairs · 2021-09-27

**Decision:**

Accept (Poster)

**Comment:**

The paper proposes a new 1-bit mean estimation algorithm. While the idea of using random rotation matrices have been explored in this problem, the paper proposes a new elegant algorithm that has better performance both in theory and experiments. The paper is well written and I recommend acceptance.

I am curious to see how this method compares empirically to the variable length encoding presented in Distributed mean estimation with limited communication paper. I encourage authors to add this comparison and incorporate other reviewer comments in the final version.